# Incomplete Multi-view Deep Clustering with Data Imputation and Alignment

**Jiyuan Liu**[1], **Xinwang Liu**[1*], **Xinhang Wan**[1], **Ke Liang**[1], **Weixuan Liang**[1]
**Sihang Zhou**[1], **Huijun Wu**[1] **and Kehua Guo**[2]
[1] National University of Defense Technology, Changsha, Hunan, China. 410072.
[2] Central South University, Changsha, Hunan, China. 410083.
`liujiyuan13@nudt.edu.cn`

## Abstract

Incomplete multi-view deep clustering is an emerging research hot-pot to incorporate data information of multiple sources or modalities when parts of them are missing. Most of existing approaches encode the available data observations into multiple view-specific latent representations and subsequently integrate them for the next clustering task. However, they ignore that the latent representations are unique to a fixed set of data samples in all views. Meanwhile, the pair-wise similarities of missing data observations are also failed to utilize in latent representation learning sufficiently, leading to unsatisfactory clustering performance. To address these issues, we propose an incomplete multi-view deep clustering method with data imputation and alignment. Assuming that each data sample corresponds to a same latent representation among all views, it projects the latent representations into feature spaces with neural networks. As a result, not only the available data observations are reconstructed, but also the missing ones can be imputed accordingly. Moreover, a linear alignment measurement of linear complexity is defined to compute the pair-wise similarities of all data observations, especially including those of the missing. By executing the above two procedures iteratively, the discriminative latent representations can be learned and used to group the data into categories with off-the-shelf clustering algorithms. In experiment, the proposed method is validated on a set of benchmark datasets and achieves state-of-the-art performances.

## 1 Introduction

With the development of electronic device and information technology, the data observations are widely accumulated from different sources and modalities, collectively referred to as multi-view data in academic communities. For instance, when making advertisement recommendation to Internet users, the content provider would collect their personal details from multiple aspects in advance, such as living habit, shopping record, etc. [1] In the diagnosis and treatment of Alzheimer's disease, multiple types of data are collected and analyzed along with a set of medical examinations, including blood test, Cerebrospinal Fluid (CSF) examination, Magnetic Resonance Imaging (MRI), Computed Tomography (CT), Positron Emission Tomography (PET), etc. [2] Therefore, how to integrate the multi-view data effectively and efficiently is a critical problem. In this background, multi-view clustering can explore the consensus and complementary information among different data views and improve the performance over single-view clustering by large margins, catching a large volume of attentions from researchers [3].

---

*Corresponding author

39th Conference on Neural Information Processing Systems (NeurIPS 2025).

Since most of the existing multi-view clustering are derived from single-view algorithms, they can be grouped into five categories accordingly, i.e., multiple kernel clustering [4, 5, 6, 5], multi-view subspace clustering [7, 8, 9], multi-view spectral clustering [10, 11], multi-view matrix factorization [12, 13] and multi-view deep clustering [14, 15]. Insides, multi-view deep clustering is emerging in recent years along with the rapid development of neural network techniques. Among the advances, Cui et al. propose to extract high-level features from data observations of each view with multiple view-specific fully connected auto-encoders, then design the dual contrasting losses to maximize the distance between different clusters and enhance the compactness within clusters [16]. Yu et al. employ graph convolutional encoder on each data view [17]. Subsequently, the contrastive learning technique and block diagonalization constraint are adopted to guide representation matrix learning, while representation learning is utilized into clustering result generation. Similarly, Chen et al. learn view-invariant data representations with auto-encoders and compute results by contrasting the cluster assignments among different views [18]. Usually, multi-view deep clustering approaches achieve better performance than the others due to the superior capability of neural networks to generate high-quality latent representations on data observations.

Due to sensor failure or restricted conditions, the collected multi-view data are often incomplete where one or more views are missing completely. In such setting, the researchers propose a number of incomplete multi-view deep clustering approaches to make the most of available data observations rather than discarding the data samples with missing views [19, 20]. Mostly, they first encode the available data observations into multiple view-specific latent representations and subsequently integrate them by designing different losses on them. For instance, Chao et al. adopt the contrastive loss to maximize representation similarities of the same data sample in different views while minimize those of different data samples [21]. At the same time, Lin et al. modify the contrastive loss to maximize mutual information across all data views, promoting the learning of informative and consistent latent representations [22].

Although the existing methods achieves satisfactory performance, they ignore the fact that the latent representations of a fixed set of data samples are unique and invariant to different views. Meanwhile, they only concentrate on the integration of available data observations but overlook the pair-wise similarity information among missing data observations, limiting the further improvement of clustering performance. To address these issues, we propose a novel Incomplete Multi-view Deep Clustering with Data Imputation and Alignment (IMDC-DIA). Specifically, it assumes that all views share a same latent representation with respect to a certain data sample. Contrary to the mainstream methods of encoding data observations into latent representations, the proposed approach projects the unique latent representations to feature space of each view with multiple independent neural networks. As a consequence, not only the available data observations are reconstructed, but also the missing ones can be imputed accordingly. Moreover, a linear alignment measurement is defined and analyzed accordingly. Then, we adopt it to align the latent representations with the data observations of each view, so as to utilize the pair-wise similarities of all data observations, especially including those of the missing. By executing the above data imputation and alignment coherently and iteratively, the discriminative latent representations can be learned and used to group the data into categories with off-the-shelf clustering algorithms. To validate effectiveness of the proposed IMDC-DIA method, we conduct extensive experiments on a set of benchmark datasets and compare it with the most recent advances in literature. Corresponding results show it achieves state-of-the-art clustering performance in almost all settings, well validating its effectiveness and superiority.

## 2 Related work

### 2.1 Multi-view deep clustering

As a representative of multi-view clustering, multi-view deep clustering is one of the most effective technique to group data samples with integrating their multi-view information. Typically, it achieves better performances than the other multi-view clustering methods, since the adopted neural networks can extract higher-quality latent representations on the data observations of each view. Among them, the Multiple Layer Perceptron (MLP) network is the most widely used in literature, such as those in [16, 23]. Besides, the other types of neural networks are also employed to further improve the quanlity of latent representations. For example, Yu et al. generate the view-specific latent representations with Graph Convolutional Network (GCN) encoder on each view, successfully capturing the structural data information along with learning attribute features [17]. Observing the poor scalability of MLP,

Zhu et al. transform the feature vector into image and use Convolutional Neural Networks (CNN) subsequently, achieving satisfactory clustering performance [24]. Yang et al. adopt the Transformer structure by utilizing self-attention mechanism, enabling the encoder to better retain complementary information across different views and remove the exclusive information [25].

Apart from the neural network structure, plenty of constraints and loss functions are developed in existing researches. Insides, contrastive loss is the most popular one of them. Commonly, it regards the latent representations of a data sample in different views as positive pairs while the rest as negative pairs, and maximizes the similarities of the former while minimizes those of the latter [25, 23]. At the same time, Cui et al. integrate the pseudo-labels in training stage and consider the latent representations of data samples in a same temporary cluster as positive pair, improving the clustering performance effectively [16]. In addition, a large number of researches employ the self-representation loss derivate from the self-representation constraint in classical multi-view clustering [24, 26]. It assumes each latent representation to be a linear sum of the others and minimizes their differences. Nevertheless, some of the other popular constraints and loss functions would be the adversarial similarity constraint [27], the discriminative constraint [15] and the self-supervised loss [17].

## 2.2 Incomplete multi-view deep clustering

To deal with incomplete data in real-world scenarios, incomplete multi-view deep clustering is emerging to be a promising approach and explored in recent literature. Some of them only rely on the available data observations to compute the clustering results [28, 29]. They usually encode the available data observations into latent representations with multiple neural networks with each in one data view. On the basis, Wen et al. propose a graph embedding strategy to simultaneously capture the high-level features and local structure of each data view, while a self-paced strategy is also utilized to select the most confident samples in model training, reducing the negative influence of outliers [29]. Differently, Xu et al. learn latent representations by incorporating the Mixture-of-Gaussians prior information to enhance their clustering-friendly structure and develop a Product-of-Experts approach to efficiently aggregate them [30]. Xu et al. utilize all view-specific latent representations into a consensus one with an adaptive feature projection module, avoiding to impute the missing data [31]. Then, the correlated common cluster information is explored by maximizing its mutual information, while the distribution alignment is achieved by minimizing its mean discrepancy coherently.

Different from the above approaches, the other incomplete multi-view deep clustering methods try to recover the missing data along with the latent representation learning and data clustering [21, 32, 33]. For instance, Wang et al. learn the low-dimensional latent representations with view-specific encoder networks and explicitly generate the missing data with generative adversarial networks at the same time [33]. Lin et al. unify the cross-view consistency learning and data recovery techniques by maximizing the mutual information of different views and minimizing the conditional entropy through dual prediction, respectively [22, 34]. Liu et al. propose a two-stage autoencoder network with recurrent graph reconstruction mechanism to extract high-level latent representations and recover the missing data synchronously [35].

## 3 Methodology

In this section, the proposed IMDC-DIA method is introduced, where its framework is visualized in Fig. 1. Briefly, it aims to learn the high-quality latent representations which are subsequently fed to off-the-shelf clustering algorithms so as to group the data accurately. To accomplish this, the latent representations are assumed to be unique to all views. On the basis, three main components are concerned, i.e., data reconstruction, imputation and alignment. On the left of Fig. 1, the proposed IMDC-DIA method projects the unique latent representations to feature space of each view with multiple independent neural networks. Thereby, not only the available data observations are reconstructed (data reconstruction), but also the missing ones can be imputed and completed (data imputation) accordingly. In data alignment, a linear alignment measurement is defined and, as shown on the right of Fig. 1, is adopted to align the unique data representations with the completed data observations of each view.

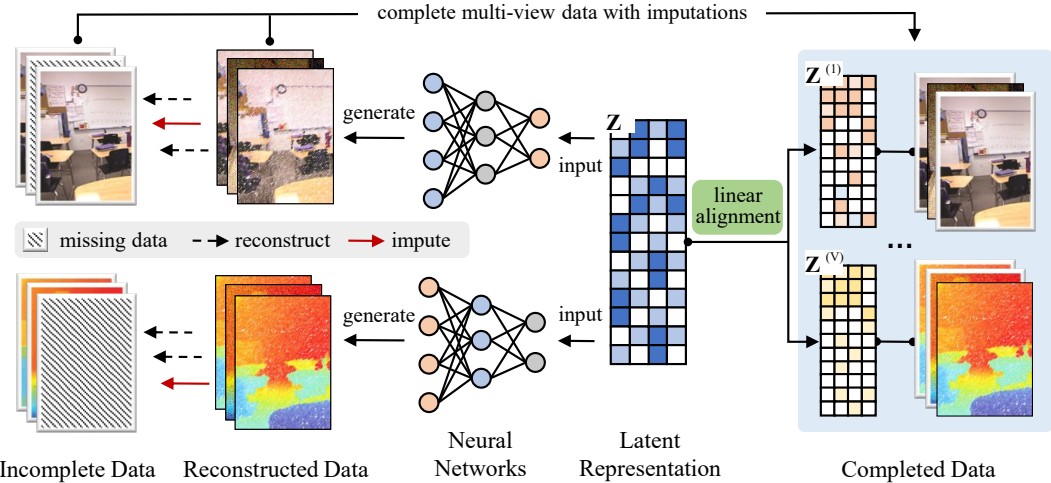

Figure 1: Framework of the proposed IMDC-DIA method. For the ease of expression, only two data views are presented, but arbitrary number of views are supported. Also, any types of neural networks, apart from the fully-connected, can be employed, only if they project the latent representations into data feature space.

## 3.1 Formulation

Denoting $N$, $V$ and $D_v$ as the number of data samples, the number of views and the feature dimension of $v$-th view, the multi-view data observations can be specified to $\{\mathbf{x}_i^{(v)}\}_{i,v=1}^{N,V}$ in which $\mathbf{x}_i^{(v)} \in \mathbb{R}^{D_v}$. Corresponding availability indicators are $\{\mathbf{a}_i\}_{i=1}^N$ where $\mathbf{a}_i$ is a discrete vector in $\{0,1\}^V$. Here, the data observation of $v$-th view $\mathbf{x}_i^{(v)}$ is available if the $v$-th element $\mathbf{a}_{i,v}$ is 1, while missing if 0. In such setting, incomplete multi-view deep clustering methods aim to group the data into clusters with the available observations where the cluster number $K$ is known in advance.

### 3.1.1 Data reconstruction and imputation

Instead of encoding the available data observations into multiple view-specific latent representations, the proposed IMDC-DIA method assumes that the data latent structure is intrinsic and corresponding latent representations should be unique to all data views. Denoting the latent representations to be $\{\mathbf{z}_i\}_{i=1}^N$ where $\mathbf{z}_i \in \mathbb{R}^D$, it proposes to reconstruct the multi-view data observations by projecting the latent representations into feature spaces with $V$ independent neural networks as

$$\hat{\mathbf{x}}_i^{(v)} = g_{\Theta_v}(\mathbf{z}_i), \tag{1}$$

where $\Theta_v$ represents parameters of the $v$-th neural network. It is worth to note that the latent representation $\mathbf{z}_i$ is not fixed but a variable which can be optimized in model training. Correspondingly, we can obtain the reconstruction loss by minimizing the differences between the generated data and the available data observations via

$$L_r = \frac{1}{N}\sum_{i=1}^N \frac{1}{V_i}\sum_{v=1}^V \mathbf{a}_{i,v}\|\hat{\mathbf{x}}_i^{(v)} - \mathbf{x}_i^{(v)}\|_2^2, \tag{2}$$

in which $V_i$ refers to the number of available views of $i$-th data sample. Nevertheless, it can be observed in aforementioned reconstruction process that not only the available data observations are reconstructed, but also the missing ones are imputed accordingly. Therefore, the multi-view data can be completed by filling the missing with those imputed in Eq. (1). As a result, the complete data can be formulated as

$$\bar{\mathbf{x}}_i^{(v)} = \begin{cases} \mathbf{x}_i^{(v)} & \text{if} \quad \mathbf{a}_{i,v} = 1, \\ \hat{\mathbf{x}}_i^{(v)} & \text{if} \quad \mathbf{a}_{i,v} = 0. \end{cases} \tag{3}$$

### 3.1.2 Data alignment

On the basis of the complete multi-view data of Eq. (3), the proposed IMDC-DIA method also incorporate the pair-wise similarities of all data observations, especially those of the missing, to regularize the latent representations. Primarily, the *Linear Alignment* of two matrices is defined in Definition 1.

**Definition 1 (Linear Alignment)** *Given two arbitrary matrices $\mathbf{X}_1$ and $\mathbf{X}_2$ of size $n \times d_1$ and $n \times d_2$ respectively, the linear alignment is computed as*

$$LA = \frac{L_F^2(\mathbf{X}_1^\top \mathbf{X}_2)}{L_F(\mathbf{X}_1^\top \mathbf{X}_1) \cdot L_F(\mathbf{X}_2^\top \mathbf{X}_2)}, \tag{4}$$

*in which $L_F(\cdot)$ denotes the Frobenius norm of matrix. Corresponding computation complexity is linear to $n$.*

Denoting vector $\mathbf{x}_i$ to the $i$-th row of an arbitrary matrix $\mathbf{X}$, the **pair-wise linear similarity** between its $i$-th and $j$-th rows is measured by their dot-products that

$$k_{i,j} = \mathbf{x}_i \mathbf{x}_j^\top, \quad w.r.t. \ i, j \in \{1, 2, \cdots, N\} \tag{5}$$

With considering two arbitrary matrices $\mathbf{X}_1$ and $\mathbf{X}_2$ coherently, their pair-wise linear similarities should be

$$k_{i,j}^1 = \mathbf{x}_i^1 \mathbf{x}_j^{1\top} \quad \text{and} \quad k_{i,j}^2 = \mathbf{x}_i^2 \mathbf{x}_j^{2\top} \tag{6}$$

On this basis, the similarity **consistency** of these two matrices can be measured by the sum of their dot-products as

$$c = \frac{\sum_{i,j=1}^N k_{i,j}^1 k_{i,j}^2}{\sqrt{\sum_{i,j=1}^N k_{i,j}^1 k_{i,j}^1} \sqrt{\sum_{i,j=1}^N k_{i,j}^2 k_{i,j}^2}} \tag{7}$$

where the denominator is a scaler term to ensure the obtained consistency in range $[-1, 1]$.

**Theorem 1** *The linear alignment of two arbitrary matrices measures the consistency of the pair-wise linear similarities of their rows.*

**Proof 1** *The consistency of Eq. (7) can be transformed to*

$$
\begin{aligned}
c &= \frac{\sum_{i,j=1}^N k_{i,j}^1 k_{i,j}^2}{\sqrt{\sum_{i,j=1}^N k_{i,j}^1 k_{i,j}^1} \sqrt{\sum_{i,j=1}^N k_{i,j}^2 k_{i,j}^2}} = \frac{\sum_{i,j=1}^N k_{i,j}^1 k_{j,i}^2}{\sqrt{\sum_{i,j=1}^N k_{i,j}^1 k_{i,j}^1} \sqrt{\sum_{i,j=1}^N k_{i,j}^2 k_{i,j}^2}} \\
&= \frac{< \mathbf{K}_1, \mathbf{K}_2 >_F}{\sqrt{< \mathbf{K}_1, \mathbf{K}_1 >_F < \mathbf{K}_2, \mathbf{K}_2 >_F}} = \frac{\mathrm{Tr}[(\mathbf{X}_1\mathbf{X}_1^\top)(\mathbf{X}_2\mathbf{X}_2^\top)]}{\sqrt{\mathrm{Tr}[(\mathbf{X}_1\mathbf{X}_1^\top)(\mathbf{X}_1\mathbf{X}_1^\top)] \cdot \mathrm{Tr}[(\mathbf{X}_2\mathbf{X}_2^\top)(\mathbf{X}_2\mathbf{X}_2^\top)]}} \\
&= \frac{\mathrm{Tr}[(\mathbf{X}_1^\top \mathbf{X}_2)^\top (\mathbf{X}_1^\top \mathbf{X}_2)]}{\sqrt{\mathrm{Tr}[(\mathbf{X}_1^\top \mathbf{X}_1)(\mathbf{X}_1^\top \mathbf{X}_1)] \cdot \mathrm{Tr}[(\mathbf{X}_2^\top \mathbf{X}_2)(\mathbf{X}_2^\top \mathbf{X}_2)]}} = \frac{\|\mathbf{X}_1^\top \mathbf{X}_2\|_F^2}{\|\mathbf{X}_1^\top \mathbf{X}_1\|_F \cdot \|\mathbf{X}_2^\top \mathbf{X}_2\|_F} = LA,
\end{aligned} \tag{8}
$$

*in which the second equation holds for*

$$k_{j,i}^2 = \mathbf{x}_{2,j} \mathbf{x}_{2,i}^\top = \mathbf{x}_{2,i} \mathbf{x}_{2,j}^\top = k_{i,j}^2, \tag{9}$$

*while the third holds for*

$$
\mathbf{K}_1 = \begin{bmatrix}
\mathbf{x}_{1,1}\mathbf{x}_{1,1}^\top & \mathbf{x}_{1,1}\mathbf{x}_{1,2}^\top & \cdots & \mathbf{x}_{1,1}\mathbf{x}_{1,N}^\top \\
\mathbf{x}_{1,2}\mathbf{x}_{1,1}^\top & \mathbf{x}_{1,2}\mathbf{x}_{1,2}^\top & \cdots & \mathbf{x}_{1,2}\mathbf{x}_{1,N}^\top \\
\vdots & \vdots & \ddots & \vdots \\
\mathbf{x}_{1,N}\mathbf{x}_{1,1}^\top & \mathbf{x}_{1,N}\mathbf{x}_{1,2}^\top & \cdots & \mathbf{x}_{1,N}\mathbf{x}_{1,N}^\top
\end{bmatrix} = \begin{bmatrix}
k_{1,1}^1 & k_{1,2}^1 & \cdots & k_{1,3}^1 \\
k_{2,2}^1 & k_{2,2}^1 & \cdots & k_{2,3}^1 \\
\cdots & \cdots & \ddots & \cdots \\
k_{N,1}^1 & k_{N,2}^1 & \cdots & k_{N,N}^1
\end{bmatrix} \quad and
$$

$$
\mathbf{K}_2 = \begin{bmatrix}
\mathbf{x}_{2,1}\mathbf{x}_{2,1}^\top & \mathbf{x}_{2,1}\mathbf{x}_{2,2}^\top & \cdots & \mathbf{x}_{2,1}\mathbf{x}_{2,N}^\top \\
\mathbf{x}_{2,2}\mathbf{x}_{2,1}^\top & \mathbf{x}_{2,2}\mathbf{x}_{2,2}^\top & \cdots & \mathbf{x}_{2,2}\mathbf{x}_{2,N}^\top \\
\vdots & \vdots & \ddots & \vdots \\
\mathbf{x}_{2,N}\mathbf{x}_{2,1}^\top & \mathbf{x}_{2,N}\mathbf{x}_{2,2}^\top & \cdots & \mathbf{x}_{2,N}\mathbf{x}_{2,N}^\top
\end{bmatrix} = \begin{bmatrix}
k_{1,1}^2 & k_{1,2}^2 & \cdots & k_{1,3}^2 \\
k_{2,2}^2 & k_{2,2}^2 & \cdots & k_{2,3}^2 \\
\cdots & \cdots & \ddots & \cdots \\
k_{N,1}^2 & k_{N,2}^2 & \cdots & k_{N,N}^2
\end{bmatrix} \tag{10}
$$

*This completes the proof.*

According to Theorem 1, the linear alignment of two arbitrary matrices measures the consistency of the pair-wise linear similarities of their rows. To consider the pair-wise similarities of all data observations, especially including those of the missing ones, we propose to maximize the linear alignment between the latent representations and the completed data observations of $v$-th view, i.e., $\max LA(\mathbf{Z}, \bar{\mathbf{X}}^{(v)})$ Therefore, corresponding loss can be formulated to

$$L_a^{(v)} = -LA(\mathbf{Z}, \bar{\mathbf{X}}^{(v)}) = -\frac{L_F^2(\mathbf{Z}^\top \bar{\mathbf{X}}^{(v)})}{L_F(\mathbf{Z}^\top \mathbf{Z}) \cdot L_F(\bar{\mathbf{X}}^{(v)\top} \bar{\mathbf{X}}^{(v)})}, \tag{11}$$

in which $\mathbf{Z}$ and $\bar{\mathbf{X}}^{(v)}$ are the matrix form of $\{\mathbf{z}_i\}_{i=1}^N$ and $\{\bar{\mathbf{x}}_i^{(v)}\}_{i=1}^N$, respectively. Furthermore, to consider the data quality of each view, an additional set of weight parameters $\{w_v\}_{v=1}^V$ are introduced to balance their effect on latent representations. As a consequence, the overall data alignment loss can be written to

$$L_a = \sum_{v=1}^V w_v L_a^{(v)}, \quad s.t. \sum_{v=1}^V w_v^2 = 1. \tag{12}$$

It is worth to note that $\{w_v\}_{v=1}^V$ are variables to be optimized in model learning rather than hyper-parameters specified in advance.

### 3.1.3 Loss function

As seen in the framework of Fig. 1, the proposed IMDC-DIA method is composed of three main components, including the data reconstruction, imputation and alignment. By utilizing the reconstruction loss of Eq. (1), missing data imputation of Eq. (3) and data alignment loss of Eq. (12), the overall loss function is written to

$$L = L_r + \beta L_a, \quad s.t. \sum_{v=1}^V w_v^2 = 1 \tag{13}$$

in which $\beta$ is a trade-off parameter and should be specified before model training.

## 3.2 Optimization

According to the overall loss of Eq. (13), there are three variables to optimize in model training, including the neural network parameter $\{\Theta_v\}_{v=1}^V$, the unique latent representation $\{\mathbf{z}_i\}_{i=1}^N$ and weight parameter $\{w_v\}_{v=1}^V$. Each of them can be optimized in the following.

*Optimization of neural network parameter* $\{\Theta_v\}_{v=1}^V$. Same to most of deep learning methods, the neural network parameter can be optimized with gradient descent strategy. In experiment, we adopt the popular Adaptive Moment Estimation (Adam) optimizer.

*Optimization of latent representation* $\{\mathbf{z}_i\}_{i=1}^N$. Different from the existing multi-view clustering methods, such as [13], the proposed IMDC-DIA method is difficult to find the close-form solution of data latent representation. Therefore, the gradient descent strategy with Adam optimizer is adopted in its optimization.

*Optimization of weight parameter* $\{w_v\}_{v=1}^V$. With fixing the others, $L_a^{(v)}$ is given and minimizing the overall loss of Eq. (13) equals to

$$\max_{\{w_v\}_{v=1}^V} \sum_{v=1}^V w_v(-L_a^{(v)}), \quad s.t. \sum_{v=1}^V w_v^2 = 1. \tag{14}$$

According to o Cauchy–Schwarz inequality [36],

$$\sum_{v=1}^V w_v(-L_a^{(v)}) \le \sqrt{\left(\sum_{v=1}^V w_v^2\right)\left(\sum_{v=1}^V L_a^{(v)2}\right)} = \sqrt{\sum_{v=1}^V L_a^{(v)2}}, \tag{15}$$

where the equality holds for when

$$\frac{w_1}{-L_a^{(1)}} = \frac{w_2}{-L_a^{(2)}} = \cdots = \frac{w_V}{-L_a^{(V)}}. \tag{16}$$

Unifying the constraint $\sum_{v=1}^{V} w_v^2 = 1$ in Eq. (14), the solution can be computed to

$$w_v^* = -L_a^{(v)} / \sqrt{\sum_{v'=1}^{V} L_a^{(v')2}}. \tag{17}$$

Overall, the aforementioned three parameters are optimized alternately and the latent representations can be achieved until the convergence or maximal epoch. Next, $k$-means algorithm is applied on the computed latent representations $\{\mathbf{z}_i\}_{i=1}^{N}$ to group multi-view data into categories. Furthermore, the pseudo-code of the optimization procedure is summarized in Alg. 1.

---

**Algorithm 1** Incomplete Multi-view Deep Clustering with Data Imputation and Alignment

---

**Input**: incomplete multi-view data $\{\mathbf{x}_i^{(v)}\}_{i,v=1}^{N,V}$ with availability indicator $\{\mathbf{a}_i\}_{i=1}^{N}$, cluster number $k$
**Output**: data labels
 1:  initialize latent representation $\{\mathbf{z}_i\}_{i=1}^{N}$ and network parameters $\{\Theta_v\}_{v=1}^{V}$;
 2:  $t = 0$;
 3:  **while** $t < epochs$ **do**
 4:      # *forward*
 5:      compute $L_r$ with Eq. (2);
 6:      complete multi-view data with Eq. (3);
 7:      compute $\{L_a^{(v)}\}_{v=1}^{V}$ with Eq. (11);
 8:      update the view weights $\{w_v\}_{v=1}^{V}$ with Eq. (17);
 9:      compute the overall loss $L$ with Eq. (13);
10:      # *back propagation*
11:      update latent representation $\{\mathbf{z}_i\}_{i=1}^{N}$ and network parameters $\{\Theta_v\}_{v=1}^{V}$;
12:      $t = t + 1$;
13:  **end while**
14:  compute the data labels with $k$-means on latent representations $\{\mathbf{z}_i\}_{i=1}^{N}$;

---

### 3.3 Computation complexity

Specifically, the data reconstruction and imputation of Eq. (1), (2) and (3) only require projecting the latent representations into feature spaces with parameterized neural networks, hence introducing a $\mathcal{O}(N)$ complexity. Also, corresponding updates on latent representations and network parameters are of $\mathcal{O}(N)$ complexity. Besides, the complexity of data alignment loss $L_a^{(v)}$ in Eq. (12) is mainly on the computation of $L_F^2(\mathbf{Z}^\top \bar{\mathbf{X}}^{(v)})$, $L_F(\mathbf{Z}^\top \mathbf{Z})$ and $L_F(\bar{\mathbf{X}}^{(v)\top} \bar{\mathbf{X}}^{(v)})$ which are all of $\mathcal{O}(N)$ complexity. Nevertheless, the optimization of weight parameter $\{w_v\}_{v=1}^{V}$ via Eq. (17) is of $\mathcal{O}(V)$ complexity. In summary, the overall complexity of the proposed IMDC-DIA method is linear to the number of data samples, i.e., $\mathcal{O}(N)$.

## 4 Experiment

### 4.1 Experiment setting

To validate the proposed IMDC-DIA method, we conduct extensive experiments on four benchmark datasets, including HandWritten[2] [37], Caltech5V[3] [38], Flower17[4] [39] and MSRCV1[5] [40]. On the basis, we generate incomplete datasets by following the common strategy [41] in literature. Specifically, assuming the missing ratio to be $m$, $m$ percent of data samples are selected to remove at least one views randomly. In the following experiments, $m$ is set in $\{0.1, 0.3, 0.5, 0.7.0.9\}$. Meanwhile, five recent incomplete multi-view deep clustering approaches are considered in comparison, including DITA-IMVC [42], DSIMVC [41], DCP [34], DVIMVC [30] and CPSPAN [43]. Note that, we

---

[2]`https://archive.ics.uci.edu/ml/datasets/Multiple+Features`
[3]`https://data.caltech.edu/records/mzrjq-6wc02`
[4]`https://www.robots.ox.ac.uk/~vgg/data/flowers/17`
[5]`https://github.com/youweiliang/Multi-view_Clustering`

Table 1: Performance comparison between the proposed IMDC-DIA and recent incomplete deep multi-view clustering approaches. The *avg.* column refers to performance averages of all missing ratios. Note that, the best results are marked in bold, while the second-best with underline.

| Metric | | ACC | | | | | | NMI | | | | |
|---|---|---|---|---|---|---|---|---|---|---|---|---|
| Missing ratio | | 0.1 | 0.3 | 0.5 | 0.7 | 0.9 | avg. | 0.1 | 0.3 | 0.5 | 0.7 | 0.9 | avg. |
| HandWritten | DITA-IMVC | 75.48 | 78.92 | 81.37 | 81.02 | 55.00 | 74.36 | 75.81 | 78.05 | 77.62 | 75.54 | 52.16 | 71.84 |
| | DSIMVC | 79.55 | 80.73 | 78.83 | 77.48 | 50.87 | 73.49 | 78.57 | 77.76 | 74.93 | 71.97 | 48.53 | 70.35 |
| | DCP | 81.95 | 75.73 | 77.23 | 71.77 | 13.07 | 63.95 | 84.37 | 78.95 | 79.22 | 74.39 | 0.93 | 63.57 |
| | DVIMC | 86.42 | 83.05 | 45.48 | 25.20 | 18.92 | 51.81 | 87.64 | 84.78 | 56.58 | 35.43 | 17.96 | 56.48 |
| | CPSPAN | 90.42 | 91.25 | 91.08 | 90.27 | 86.73 | 89.95 | 83.84 | 84.37 | 84.01 | 83.55 | 82.62 | 83.68 |
| | IMDC-DIA | **96.37** | **93.68** | **91.80** | **90.65** | **87.93** | **92.09** | **91.80** | **93.68** | **91.80** | **90.65** | **87.93** | **91.17** |
| Caltech5V | DITA-IMVC | 79.10 | 75.76 | 68.02 | 58.90 | 40.29 | 64.41 | 67.56 | 64.78 | 58.84 | 52.40 | 29.29 | 54.57 |
| | DSIMVC | 76.64 | 73.14 | 66.36 | 57.38 | 44.95 | 63.69 | 68.02 | 63.11 | 56.71 | 49.23 | 33.99 | 54.21 |
| | DCP | 44.50 | 46.95 | 46.45 | 44.67 | 16.98 | 39.91 | 45.22 | 52.31 | 50.69 | 45.07 | 0.54 | 38.77 |
| | DVIMC | **88.64** | 81.36 | **84.93** | 80.86 | 68.81 | 80.92 | 80.52 | 73.38 | 76.27 | 73.56 | 62.40 | 73.23 |
| | CPSPAN | 83.29 | 81.10 | 77.21 | 76.79 | 79.02 | 79.48 | 73.81 | 71.30 | 68.73 | 68.43 | 69.26 | 70.31 |
| | IMDC-DIA | 86.05 | **85.05** | 83.33 | **85.02** | **81.29** | **84.15** | **86.05** | **85.05** | **83.33** | **85.02** | **81.29** | **84.15** |
| Flower17 | DITA-IMVC | 18.75 | 17.52 | 13.77 | 13.16 | 12.84 | 15.21 | 18.15 | 16.60 | 10.23 | 8.65 | 8.66 | 12.46 |
| | DSIMVC | 18.63 | 16.86 | 13.50 | 13.36 | 13.36 | 15.14 | 17.51 | 14.78 | 9.29 | 9.27 | 9.22 | 12.01 |
| | DCP | 26.15 | 25.49 | 22.21 | 15.05 | 10.78 | 19.94 | 26.18 | 25.94 | 23.47 | 15.37 | 3.69 | 18.93 |
| | DVIMC | 36.69 | 31.23 | 26.84 | 23.41 | 21.69 | 27.97 | 35.43 | 29.82 | 24.30 | 20.93 | 20.08 | 26.11 |
| | CPSPAN | 36.27 | 39.34 | 31.15 | 37.35 | 36.00 | 36.02 | 37.13 | 39.65 | 30.88 | 38.15 | 35.97 | 36.36 |
| | IMDC-DIA | **52.03** | **46.47** | **44.22** | **41.54** | **39.14** | **44.68** | **52.03** | **46.47** | **44.22** | **41.54** | **39.14** | **44.68** |
| MSRCV1 | DITA-IMVC | 76.03 | 74.60 | 70.95 | 66.83 | 55.87 | 68.86 | 67.62 | 63.02 | 60.61 | 54.96 | 43.32 | 57.91 |
| | DSIMVC | 78.57 | 76.98 | 70.48 | 71.59 | 64.76 | 72.48 | 68.37 | 65.81 | 62.26 | 59.19 | 51.78 | 61.48 |
| | DCP | 17.94 | 20.00 | 22.06 | 22.22 | 22.70 | 20.98 | 7.11 | 7.54 | 7.05 | 4.95 | 4.62 | 6.25 |
| | DVIMC | 74.60 | 61.43 | 56.67 | 43.17 | 38.25 | 54.82 | 66.68 | 56.40 | 54.91 | 42.02 | 30.08 | 50.02 |
| | CPSPAN | 85.08 | 86.51 | 85.40 | 84.29 | 77.62 | 83.78 | 75.86 | 77.25 | 75.51 | 74.09 | 69.06 | 74.35 |
| | IMDC-DIA | **91.27** | **92.22** | **86.03** | **85.08** | **83.02** | **87.52** | **91.27** | **92.22** | **86.03** | **85.08** | **83.02** | **87.52** |

directly adopt the public codes available on the authors' websites without modification. To ensure the comparison fairness, we grid-search their parameters and report the best. So does the proposed IMDC-DIA[6] method with setting its only parameter $\beta$ in $\{0.01, 0.1, 1, 10, 100\}$. By following the literature, four most common metrics are adopted to measure the clustering performance, including accuracy (ACC), normalized mutual information (NMI), purity (PUR) and adjusted rand index (ARI). Nevertheless, all methods are executed multiple times and their averages are reported to remove randomness effect.

## 4.2 Effectiveness

In order to validate effectiveness of the proposed IMDC-DIA method, we compare it with the recent advances on incomplete deep multi-view clustering in literature. Corresponding results can be found in Table 1. It is obvious that the proposed IMDC-DIA outperforms the recent advances in almost all missing ratios on all datasets. Concretely, it achieves the improvements of 5.95%, 2.43%, 0.72%, 0.38%, 1.20% in accuracy and 4.16%, 8.90%, 7.79%, 7.10%, 5.31% in NMI on HandWritten; 15.34%, 7.13%, 13.07%, 4.19%, 3.14% in accuracy and 14.90%, 6.82%, 13.34%, 3.39%, 3.17% in NMI on Flower17; 6.19%, 5.71%, 0.63%, 0.79%, 5.40% in accuracy and 15.41%, 14.97%, 10.52%, 10.99%, 13.96% in NMI, respectively. Although slight decreases in 0.1 and 0.3 missing ratios on Caltech5V are observed, i.e., 2.59% and 1.60%, the proposed IMDC-DIA obtains better performances on the other missing ratios consistently. Nevertheless, with averaging the results in all missing ratios, we can see that it improves the accuracy by 2.14%, 2.27%, 3.14%, 3.74% and the NMI by 7.49%, 10.92%, 8.32%, 13.17% on the four benchmark datasets. To be summarized, the aforementioned observations

---

[6]The code is available at `https://github.com/liujiyuan13/IMDC-DIA-code_release`.

Table 2: Performance comparison between the proposed IMDC-DIA method and its variants of modifying and removing the data imputation and alignment components. The *avg.* column refers to performance averages of all missing ratios, while the *avg.* row reports the performances in the setting of completing missing data with average values. Note that, the best results are marked in bold, while the second-best with underline.

| Metric | | | ACC | | | | | | NMI | | | | | |
|---|---|---|---|---|---|---|---|---|---|---|---|---|---|---|
| Missing ratio | | | 0.1 | 0.3 | 0.5 | 0.7 | 0.9 | avg. | 0.1 | 0.3 | 0.5 | 0.7 | 0.9 | avg. |
| HandWritten | w/o imp. | avg. | 75.58 | 66.17 | 56.47 | 50.63 | 41.78 | 58.13 | 75.30 | 64.92 | 55.95 | 51.55 | 42.18 | 57.98 |
| | | zero | 83.85 | 68.87 | 61.33 | 48.23 | 44.72 | 61.40 | 76.90 | 64.85 | 56.41 | 49.60 | 42.58 | 58.07 |
| | | rand | 79.98 | 69.73 | 58.37 | 55.05 | 44.33 | 61.49 | 78.33 | 68.31 | 57.06 | 52.81 | 42.45 | 59.79 |
| | | none | 96.13 | 82.55 | 87.82 | 78.70 | 55.83 | 80.21 | 91.37 | 82.39 | 82.03 | 74.80 | 49.31 | 75.98 |
| | w/o align. | | 93.62 | 86.65 | 90.57 | 81.27 | 82.78 | 86.98 | 87.78 | 81.89 | 81.82 | 76.88 | 73.44 | 80.36 |
| | IMDC-DIA | | **96.37** | **93.68** | **91.80** | **90.65** | **87.93** | **92.09** | **91.80** | **86.68** | **83.31** | **82.05** | **77.39** | **84.25** |
| Caltech5V | w/o imp. | avg. | 74.14 | 73.19 | 63.86 | 52.07 | 46.98 | 62.05 | 72.26 | 66.11 | 56.68 | 47.16 | 39.89 | 56.42 |
| | | zero | 80.74 | 73.60 | 67.38 | 61.81 | 52.12 | 67.13 | 74.06 | 63.71 | 57.33 | 52.74 | 43.37 | 58.24 |
| | | rand | 78.76 | 68.31 | 62.05 | 58.10 | 47.95 | 63.03 | 72.70 | 60.89 | 53.63 | 50.37 | 37.17 | 54.95 |
| | | none | **87.98** | 82.14 | 81.71 | 76.31 | 61.74 | 77.98 | **78.57** | 72.57 | 70.35 | 62.81 | 46.61 | 66.18 |
| | w/o align. | | 80.21 | 81.90 | 80.07 | 83.50 | **81.60** | 81.46 | 73.42 | 72.84 | 71.04 | 72.08 | 67.51 | 71.38 |
| | IMDC-DIA | | 86.05 | **85.05** | **83.33** | **85.02** | 81.29 | **84.15** | 76.39 | **75.28** | **72.22** | **72.28** | **68.14** | **72.86** |
| Flower17 | w/o imp. | avg. | 47.23 | 39.80 | 32.70 | 29.78 | 25.02 | 34.91 | 46.18 | 38.56 | 33.21 | 27.93 | 24.29 | 34.03 |
| | | zero | 45.15 | 36.86 | 31.81 | 27.38 | 24.63 | 33.17 | 44.51 | 34.58 | 29.05 | 25.27 | 22.77 | 31.24 |
| | | rand | 45.64 | 35.20 | 30.15 | 25.98 | 23.75 | 32.14 | 45.17 | 33.54 | 27.65 | 23.60 | 20.75 | 30.14 |
| | | none | 42.01 | 35.93 | 25.64 | 13.11 | 17.23 | 26.78 | 42.74 | 34.80 | 22.81 | 7.58 | 11.43 | 23.87 |
| | w/o align. | | 38.28 | 36.18 | 35.54 | 30.81 | 28.50 | 33.86 | 35.90 | 35.13 | 31.87 | 27.53 | 24.44 | 30.97 |
| | IMDC-DIA | | **51.13** | **45.00** | **42.89** | **40.69** | **37.06** | **43.35** | **47.15** | **45.02** | **41.33** | **38.42** | **34.81** | **41.35** |
| MRSCV1 | w/o imp. | avg. | 72.70 | 57.46 | 51.27 | 42.86 | 36.67 | 52.19 | 64.86 | 51.51 | 43.94 | 38.28 | 30.14 | 45.75 |
| | | zero | 68.10 | 62.86 | 56.83 | 45.56 | 42.38 | 55.14 | 59.26 | 46.69 | 46.66 | 36.53 | 30.58 | 43.94 |
| | | rand | 66.35 | 55.08 | 48.89 | 44.60 | 43.17 | 51.62 | 59.30 | 44.20 | 40.67 | 32.61 | 31.49 | 41.65 |
| | | none | 78.89 | 62.06 | 58.25 | 31.90 | 39.84 | 54.19 | 75.10 | 58.82 | 50.66 | 17.53 | 23.40 | 45.10 |
| | w/o align. | | 85.56 | 85.24 | 73.65 | 67.30 | 60.95 | 74.54 | 78.09 | 76.37 | 60.14 | 52.43 | 48.95 | 63.20 |
| | IMDC-DIA | | **91.27** | **92.22** | **86.03** | **85.08** | **83.02** | **87.52** | **84.37** | **85.63** | **77.84** | **74.21** | **70.47** | **78.51** |

well illustrate the effectiveness of the proposed IMDC-DIA method and its superiority over the recent advances in literature.

## 4.3 Ablation study

To demonstrate the rationality of our motivations and effectiveness of the proposed techniques, we further conduct an thorough ablation study in the following. Two settings are designed by modifying and removing the data imputation and alignment components. Specifically,

1. The *avg.*, *zero* and *rand* of *w/o imp.* substitute the data imputation into completing the missing data with averages of available data observations, zeros and random values, respectively. Meanwhile, the *none* of *w/o imp.* refers to only using the available data observations in data alignment module rather than completing the missing.

2. *w/o align.* removes the data alignment module, which can be easily implemented by setting parameter $\beta$ to 0.

By comparing results of the proposed IMDC-DIA method and its *w/o imp.* settings, the former achieves better clustering performances in almost missing ratios on all benchmark datasets. Although slight decreases in missing ratio 0.1 on Caltech5V are observed, i.e., 1.93% accuracy and 2.18% NMI, the proposed IMDC-DIA obtains better performances on the other missing ratios, i.e., 2.91%, 1.62%, 8.71%, 19.55% accuracy and 2.71%, 1.87%, 9.47% 21.53% NMI. In average, it improve the accuracy by 11.88%, 6.17%, 8.44%, 32.38% and the NMI by 8.27%, 6.68%, 7.32%, 32.76%. These observations well validate effectiveness of the proposed data imputation module. In addition, we

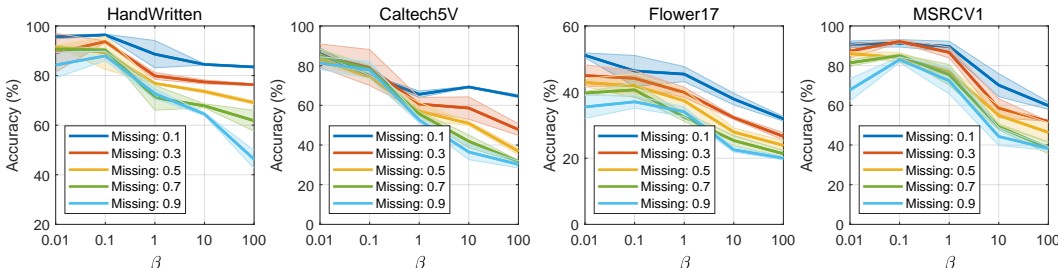

Figure 2: Accuracy variation with respect to parameter $\beta$ in different missing ratios on the four benchmark datasets. The shaded area represents the accuracy variance correspondingly.

find the proposed IMDC-DIA exhibits larger and larger performance improvements when increasing the missing ratio. Also, the results in *none* setting are better than those in *avg.*, *zero* and *rand* settings in most cases. The two observations indicate that complementing the missing data without data-irrelevant values would introduce extra bias hence decreasing the performance.

By comparing results of the proposed IMDC-DIA method and its *w/o align.* settings, similar observations can be obtained. Specifically, 0.31% accuracy decrease is presented in missing ratio 0.9 on Caltech5V, while different degrees of improvements are observed in the other missing ratios, i.e., 5.84%, 3.15%, 3.26%, 1.52% accuracy and 2.97%, 2.44%, 1.18%, 0.20% NMI. Also, with averaging the results in all missing ratios, the proposed IMDC-DIA method achieves better accuracies by 5.11%, 2.69%, 9.49% 12.98% and NMIs by 3.89%, 1.48%, 10.38%, 15.31%, respectively. These well demonstrate that the defined linear alignment is conducive to the performance improvement.

Globally, the proposed IMDC-DIA method achieves better clustering performances than its *none* of *w/o imp.* setting, i.e., 11.88%, 6.17%, 16.57%, 33.33% accuracies and 8.27%, 6.68%, 17.48%, 33.41% NMIs. This proves that incorporating the data similarities among missing data observations can largely enhance the learning of latent representations, hence obtaining better clustering results.

### 4.4 Parameter analysis

In the proposed IMDC-DIA method, the only parameter is the trade-off $\beta$ between the data reconstruction loss and data alignment loss. To investigate its effect on performance, we collect the clustering accuracies when $\beta$ set in range $\{0.01, 0.1, 1, 10, 100\}$ and present their average curves and corresponding variances in Fig. 2. In general, the clustering accuracy decreases with a larger $\beta$ in almost all missing ratios on all benchmark datasets. Specifically, it can be seen that the best accuracies are obtained mostly when setting $\beta$ to 0.01 and sometimes when setting $\beta$ to 0.1. Therefore, we recommend to set it from 0.01 to 0.1 in first priority.

## 5 Conclusion

In literature, existing incomplete multi-view deep clustering approaches overlook the fact that the latent representations should be unique for a specific set of data samples. Also, they fail to utilize the pair-wise similarities of missing data observations sufficiently. Therefore, this paper proposes an incomplete multi-view deep clustering method with data imputation and alignment by assuming each data sample corresponds to a same latent representation among all views. Insides, a novel linear alignment measurement of linear complexity is defined and incorporated to integrate the pair-wise similarities of all data observations, including those of the missing ones. Nevertheless, an alternate optimization strategy is proposed to learn high-quality latent representations for the next clustering task. In experiment, the proposed method is compared with recent advances, while its motivations and method design are validated. Also, its parameter effect is explored empirically.

Although the proposed method achieves state-of-the-art results compared to recent advances, it may be partially limited by lacking sufficient constraints on the latent representations, preventing from further improvement of the clustering performance. In the future work, we will explore more feasible constraints. Meanwhile, the evaluation on missing data imputation is also a promising research direction on the basis of this work.

## Acknowledgments and Disclosure of Funding

This work is supported by the National Natural Science Foundation of China (No. 62306324, 62376279, U24A20333, 62325604, 62276271, 62421002), the Science and Technology Innovation Program of Hunan Province (No. 2024RC3128) and the National University of Defense Technology Research Foundation (No. ZK24-30).

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

## A  Convergence analysis

In the optimization, the proposed IMDC-DIA method aims to minimize the loss $L$ of Eq. (13). Insides, three variables are optimized alternately, including the neural network parameter $\{\Theta_v\}_{v=1}^{V}$, the unique latent representation $\{\mathbf{z}_i\}_{i=1}^{N}$ and weight parameter $\{w_v\}_{v=1}^{V}$. With a little abuse of notation, let the loss be represented by

$$L = \ell(\{\Theta_v\}_{v=1}^{V}, \{\mathbf{z}_i\}_{i=1}^{N}, \{w_v\}_{v=1}^{V}). \tag{18}$$

Taking the loss at $t$ epoch to be

$$L^{(t)} = \ell(\{\Theta_v^{(t)}\}_{v=1}^{V}, \{\mathbf{z}_i^{(t)}\}_{i=1}^{N}, \{w_v^{(t)}\}_{v=1}^{V}), \tag{19}$$

By fixing $\{\mathbf{z}_i\}_{i=1}^{N}$ and $\{w_v\}_{v=1}^{V}$ at $\{\mathbf{z}_i^{(t)}\}_{i=1}^{N}$ and $\{w_v^{(t)}\}_{v=1}^{V}$, optimizing $\{\Theta_v\}_{v=1}^{V}$ at $t+1$ epoch induces to

$$\ell(\{\Theta_v^{(t)}\}_{v=1}^{V}, \{\mathbf{z}_i^{(t)}\}_{i=1}^{N}, \{w_v^{(t)}\}_{v=1}^{V}) \leq \ell(\{\Theta_v^{(t+1)}\}_{v=1}^{V}, \{\mathbf{z}_i^{(t)}\}_{i=1}^{N}, \{w_v^{(t)}\}_{v=1}^{V}). \tag{20}$$

So do the optimizations of $\{\mathbf{z}_i\}_{i=1}^{N}$ and $\{w_v\}_{v=1}^{V}$, resulting in

$$\ell(\{\Theta_v^{(t)}\}_{v=1}^{V}, \{\mathbf{z}_i^{(t)}\}_{i=1}^{N}, \{w_v^{(t)}\}_{v=1}^{V}) \leq \ell(\{\Theta_v^{(t+1)}\}_{v=1}^{V}, \{\mathbf{z}_i^{(t)}\}_{i=1}^{N}, \{w_v^{(t)}\}_{v=1}^{V})$$
$$\leq \ell(\{\Theta_v^{(t+1)}\}_{v=1}^{V}, \{\mathbf{z}_i^{(t+1)}\}_{i=1}^{N}, \{w_v^{(t)}\}_{v=1}^{V}) \leq \ell(\{\Theta_v^{(t+1)}\}_{v=1}^{V}, \{\mathbf{z}_i^{(t+1)}\}_{i=1}^{N}, \{w_v^{(t+1)}\}_{v=1}^{V}), \tag{21}$$

which can be simplified to

$$L^{(t)} \leq L^{(t+1)}. \tag{22}$$

Nevertheless,

$$L = L_r + \beta L_a \geq 0 + (-1) = -1. \tag{23}$$

To be summarized, Eq. (22) indicates that the loss monotonically decreases, while Eq. (23) illustrates that the loss is lower bounded. Therefore, the optimization algorithm is convergent.

## B  Dataset detail

The specifics of the used benchmark datasets are presented in Table 3. Also, they are briefly introduced as follows:

1. HandWritten[7] [37]. It consists of 2,000 digital images of ten classes from 0 to 9 in which there are 200 images in each class. Here, six features are used, i.e., 216-D profile correlations, 76-D Fourier coefficients of the character shapes, 64-D Karhunen-Love coefficients, 6-D morphological features, 240-D pixel averages in 2×3 windows and 47-D Zernike moments.

2. Caltech5V[8] [38]. It collects 1400 object images belonging to 7 classes. I experiment, five well-known conventional feature descriptors are adopted, including 48-D wavelet moments, 40-D CENTRIST, 254-D LBP, 512-D GIST and 928-D HOG, respectively.

3. Flower17[9] [39]. It collects flower images of 17 categories with 80 images for each class. Corresponding features are 5376-D Color Histogram, 512-D GIST, 5376-D HOG (2×2), 5376-D HOG (3×3), 1239-D LBP, 5376-D SIFT and 5376-D SSIM features.

4. MSRCV1[10] [40]. It collects 210 images of seven categories with each category provides 30 images. For each image, six features are extracted, including 1302-D CENTRIST, 48-D Color Moment, 512-D GIST, 100-D HOG, 256-D LBP and 210 SIFT, respectively.

For HandWritten, Flower17 and MSRCV1, the creators share them publicly but do not issue licenses, while Caltech5V is issued the *Creative Commons Attribution 4.0 International (CC BY 4.0)* license.

## C  Incomplete multi-view data setting

The incomplete multi-view data are generated by following the common strategy [41] in literature. Specifically, assuming the missing ratio to be $m$, $m$ percent of data samples are selected to remove at least one views randomly. In the following experiments, $m$ is set in $\{0.1, 0.3, 0.5, 0.7. 0.9\}$. Meanwhile, the corresponding psuedo-code is outlined in Alg. 2.

---

[7] https://archive.ics.uci.edu/ml/datasets/Multiple+Features

[8] https://data.caltech.edu/records/mzrjq-6wc02

[9] https://www.robots.ox.ac.uk/~vgg/data/flowers/17

[10] https://github.com/youweiliang/Multi-view_Clustering

Table 3: Details of the used benchmark datasets. Note that, in brackets are the feature dimensions of each data view.

| Dataset | Samples | Clusters | Number of Views (Dimensions) |
|---|---|---|---|
| HandWritten | 2000 | 10 | 6 (216/76/64/6/240/47) |
| Caltech5V | 1400 | 7 | 5 (48/40/254/512/928) |
| Flower17 | 1360 | 17 | 7 (512/5376/5376/1239/5376/5376) |
| MSRCV1 | 210 | 7 | 6 (1032/48/512/100/256/210) |

---

**Algorithm 2** Generation of Incomplete Multi-view Data

---

**Input**: incomplete multi-view data $\{\mathbf{x}_i^{(v)}\}_{i,v=1}^{N,V}$, missing ratio $m$
**Output**: availability indicator $\{\mathbf{a}_i\}_{i=1}^{N}$

1: obtain the numbers of data samples and views, i.e. $N$ and $V$;
2: random sample incomplete data index $\mathcal{S}$ with missing ratio $m$;
3: initialize availability indicator $\mathbf{a}_i = \{1, 2, \cdots, V\}$ w.r.t. $i \in \{1, 2, \cdots, N\}$;
4: **for** $p \in \mathcal{S}$ **do**
5:    random sample missing index $\mathbf{m}_p \subset \{1, 2, \cdots, V\}$ and $\mathbf{m}_p = \emptyset$;
6:    set availability indicator $\mathbf{a}_p = \{1, 2, \cdots, V\} - \mathbf{m}_p$;
7: **end for**

---

## D   More experiment settings

In the experiments, we adopt fully-connected neural networks on all datasets where different numbers of neurons are adopted according to different feature dimensions, as shown in Table 4. Meanwhile, the unique data latent representation and parameters of neural netowrks are both optimized with gradient descent strategy with Adam optimizer whose learning rate is set to 0.001. Additionally, the codes of the proposed method and recent advances in comparison are executed on a server with 40 *Intel(R) Xeon(R) Silver 4210R CPU @ 2.40GHz* CPUs and 2 *NVIDIA GeForce RTX 3090* GPUs. The software environment includes *Python 3.10.13* and *PyTorch 1.12.1* optimized with *CUDA 11.4*.

## E   Additional result

Apart from the ACC and NMI results in Table 1 and 2, we also provide the experiment results in PUR and ARI metrics in Table 5 and 6, respectively. It can be observed that the PUR and ARI results follow similar trends with those in ACC and NMI. Nevertheless, to analyze the training process more deeply, we also record the loss value and corresponding performances at each training epoch, which are visualized in Fig. 3. It is obvious that the loss value continuously decreases and finally converges to the minimum, while the ACC and NMI increase to the top and fluctuate around their top values. These observations well illustrate the effectiveness of loss design in Eq. (13) and the proposed IMDC-DIA method.

Table 4: The neuron numbers of the proposed IMDC-DIA method specific to different datasets.

|  | HandWritten | Caltech5V | Flower17 | MSRCV1 |
|---|---|---|---|---|
| 1st-view | (60, 64, 216) | (21, 16, 40) | (34, 512, 5376) | (21, 256, 1302) |
| 2nd-view | (60, 64, 76) | (21, 64, 254) | (34, 256, 512) | (21, 32, 48) |
| 3rd-view | (60, 32, 64) | (21, 128, 928) | (34, 512, 5376) | (21, 64, 512) |
| 4th-view | (60, 16, 6) | (21, 128, 512) | (34, 512, 5376) | (21, 32, 100) |
| 5th-view | (60, 64, 240) | (21, 256, 1984) | (34, 256, 1239) | (21, 64, 256) |
| 6th-view | (40, 32, 47) | - | (34, 512, 5376) | (21, 64, 210) |
| 7th-view | - | - | (34, 512, 5376) | - |

Table 5: Performance (PUR and ARI) comparison between the proposed IMDC-DIA and recent incomplete deep multi-view clustering approaches. The *avg.* column refers to performance averages of all missing ratios. Note that, the best results are marked in bold, while the second-best with underline.

| Metric | | PUR | | | | | | ARI | | | | | |
|---|---|---|---|---|---|---|---|---|---|---|---|---|---|
| Missing ratio | | 0.1 | 0.3 | 0.5 | 0.7 | 0.9 | avg. | 0.1 | 0.3 | 0.5 | 0.7 | 0.9 | avg. |
| HandWritten | DITA-IMVC | 77.85 | 80.90 | 81.38 | 81.02 | 55.00 | 75.23 | 66.61 | 70.78 | 70.87 | 68.67 | 35.92 | 62.57 |
| | DSIMVC | 81.13 | 80.73 | 79.88 | 77.50 | 51.18 | 74.08 | 71.69 | 70.19 | 67.27 | 63.91 | 32.36 | 61.08 |
| | DCP | 83.97 | 78.08 | 79.40 | 73.40 | 13.42 | 65.65 | 75.32 | 61.27 | 62.61 | 53.62 | 0.02 | 50.57 |
| | DVIMC | 86.53 | 83.12 | 45.53 | 25.27 | 19.45 | 51.98 | 82.38 | 77.91 | 35.65 | 14.20 | 5.74 | 43.18 |
| | CPSPAN | 90.42 | 91.25 | 91.08 | 90.27 | 87.82 | 90.17 | 80.83 | 82.12 | 81.69 | 80.46 | 78.02 | 80.62 |
| | IMDC-DIA | 96.37 | 93.68 | 91.80 | 90.65 | 87.93 | 92.09 | 92.09 | 93.68 | 91.80 | 90.65 | 87.93 | 91.23 |
| Caltech5V | DITA-IMVC | 79.10 | 75.76 | 68.02 | 60.64 | 41.31 | 64.97 | 61.06 | 57.26 | 48.75 | 40.62 | 18.10 | 45.16 |
| | DSIMVC | 76.64 | 73.14 | 66.36 | 58.57 | 46.31 | 64.20 | 60.87 | 54.83 | 47.05 | 37.67 | 22.88 | 44.66 |
| | DCP | 49.24 | 54.31 | 51.52 | 48.40 | 17.29 | 44.15 | 23.18 | 27.54 | 28.11 | 20.46 | -0.06 | 19.85 |
| | DVIMC | 88.64 | 81.36 | 84.93 | 81.38 | 68.83 | 81.03 | 77.33 | 67.12 | 71.87 | 68.58 | 52.92 | 67.56 |
| | CPSPAN | 83.29 | 81.10 | 78.14 | 77.76 | 80.05 | 80.07 | 69.41 | 65.71 | 62.80 | 61.27 | 63.60 | 64.56 |
| | IMDC-DIA | 86.05 | 85.05 | 83.33 | 85.02 | 81.29 | 84.15 | 86.05 | 85.05 | 83.33 | 85.02 | 81.29 | 84.15 |
| Flower17 | DITA-IMVC | 20.61 | 19.53 | 15.10 | 13.80 | 13.41 | 16.49 | 5.63 | 4.81 | 2.20 | 2.10 | 1.96 | 3.34 |
| | DSIMVC | 20.07 | 18.58 | 14.34 | 14.00 | 13.82 | 16.16 | 5.38 | 4.30 | 1.98 | 2.06 | 2.48 | 3.24 |
| | DCP | 27.11 | 27.18 | 23.87 | 15.76 | 11.10 | 21.00 | 10.07 | 8.02 | 3.86 | 1.76 | 0.03 | 4.75 |
| | DVIMC | 37.84 | 32.57 | 27.87 | 24.53 | 22.97 | 29.16 | 18.88 | 14.32 | 10.93 | 8.47 | 7.71 | 12.06 |
| | CPSPAN | 38.11 | 40.86 | 32.55 | 39.68 | 37.23 | 37.69 | 17.47 | 19.36 | 13.19 | 19.06 | 16.89 | 17.19 |
| | IMDC-DIA | 52.03 | 46.47 | 44.22 | 41.54 | 39.14 | 44.68 | 52.03 | 46.47 | 44.22 | 41.54 | 39.14 | 44.68 |
| MSRCV1 | DITA-IMVC | 76.83 | 74.60 | 70.95 | 66.83 | 56.67 | 69.18 | 58.42 | 54.43 | 49.80 | 43.22 | 29.20 | 47.01 |
| | DSIMVC | 78.57 | 76.98 | 70.79 | 71.59 | 65.40 | 72.67 | 59.93 | 57.68 | 51.54 | 50.56 | 41.58 | 52.26 |
| | DCP | 18.10 | 20.63 | 22.54 | 22.54 | 23.17 | 21.40 | -0.01 | 0.07 | 0.24 | -0.16 | -0.21 | -0.01 |
| | DVIMC | 74.60 | 61.43 | 56.83 | 43.81 | 38.25 | 54.98 | 57.67 | 45.83 | 41.15 | 27.52 | 19.05 | 38.24 |
| | CPSPAN | 85.08 | 86.51 | 85.40 | 84.29 | 78.10 | 83.88 | 70.63 | 72.54 | 70.95 | 69.02 | 61.63 | 68.95 |
| | IMDC-DIA | 91.27 | 92.22 | 86.03 | 85.08 | 83.02 | 87.52 | 91.27 | 92.22 | 86.03 | 85.08 | 83.02 | 87.52 |

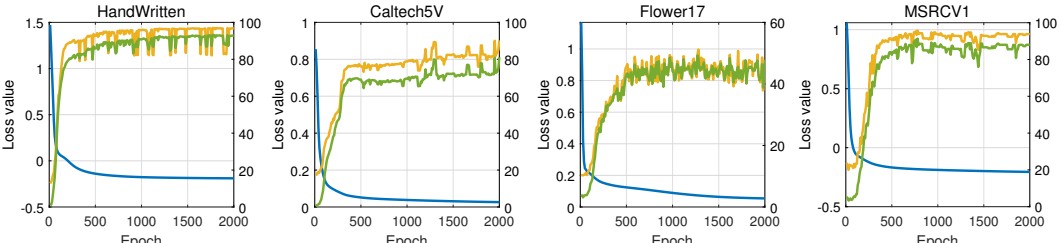

Figure 3: Loss and performance variation at each epoch on the four benchmark datasets. The blue curve represents loss value, while the yellow and green curves are accuracy and NMI, respectively.

Table 6: Performance (PUR and ARI) comparison between the proposed IMDC-DIA method and its variants of modifying and removing the data imputation and alignment components. The *avg.* column refers to performance averages of all missing ratios, while the *avg.* row reports the performances in the setting of completing missing data with average values. Note that, the best results are marked in bold, while the second-best with underline.

| Metric | | | PUR | | | | | | ARI | | | | | |
|---|---|---|---|---|---|---|---|---|---|---|---|---|---|---|
| Missing ratio | | | 0.1 | 0.3 | 0.5 | 0.7 | 0.9 | avg. | 0.1 | 0.3 | 0.5 | 0.7 | 0.9 | avg. |
| HandWritten | w/o imp. | avg | 78.12 | 68.13 | 59.27 | 53.78 | 44.48 | 60.76 | 62.70 | 44.29 | 24.49 | 18.14 | 11.49 | 32.22 |
| | | zero | 83.98 | 70.98 | 62.73 | 51.35 | 46.63 | 63.14 | 71.39 | 47.75 | 32.75 | 21.58 | 15.48 | 37.79 |
| | | rand | 80.18 | 71.13 | 59.95 | 55.62 | 46.58 | 62.69 | 70.79 | 55.61 | 32.89 | 25.63 | 15.86 | 40.16 |
| | | none | 96.13 | 84.73 | 89.28 | 79.67 | 56.68 | 81.30 | 91.59 | 77.60 | 80.11 | 69.16 | 37.18 | 71.13 |
| | w/o align. | | 93.62 | 88.18 | 90.57 | 83.77 | 84.12 | 88.05 | 86.73 | 78.23 | 80.55 | 71.71 | 69.92 | 77.43 |
| | IMDC-DIA | | **96.37** | **93.68** | **91.80** | **90.65** | **87.93** | **92.09** | **92.09** | **86.37** | **82.60** | **80.53** | **75.42** | **83.40** |
| Caltech5V | w/o imp. | avg | 78.50 | 74.48 | 64.81 | 54.62 | 48.86 | 64.25 | 60.29 | 46.02 | 30.32 | 17.49 | 12.95 | 33.41 |
| | | zero | 82.17 | 73.60 | 67.88 | 61.81 | 53.00 | 67.69 | 65.43 | 50.59 | 38.09 | 27.57 | 16.36 | 39.61 |
| | | rand | 79.24 | 68.67 | 62.43 | 58.93 | 48.24 | 63.50 | 67.77 | 51.12 | 37.88 | 28.01 | 17.16 | 40.39 |
| | | none | **87.98** | 82.45 | 81.71 | 76.31 | 61.93 | 78.08 | **76.63** | 68.91 | 68.49 | 58.79 | 38.25 | 62.22 |
| | w/o align. | | 80.81 | 82.19 | 80.93 | 83.50 | 81.60 | 81.80 | 68.21 | 68.29 | 67.07 | 68.89 | 65.25 | 67.54 |
| | IMDC-DIA | | 86.05 | **85.05** | **83.33** | **85.02** | **81.29** | **84.15** | 73.54 | **72.05** | **69.00** | **71.06** | **65.78** | **70.29** |
| Flower17 | w/o imp. | avg | 48.70 | 41.72 | 34.07 | 30.59 | 26.20 | 36.25 | 25.93 | 14.78 | 11.40 | 8.11 | 5.40 | 13.12 |
| | | zero | 45.81 | 37.55 | 32.70 | 27.97 | 25.42 | 33.89 | 27.61 | 19.97 | 13.97 | 9.85 | 7.61 | 15.80 |
| | | rand | 46.37 | 35.71 | 30.96 | 26.79 | 24.46 | 32.86 | **28.77** | 18.25 | 12.66 | 9.27 | 6.30 | 15.05 |
| | | none | 42.92 | 36.84 | 26.62 | 13.85 | 18.14 | 27.67 | 22.89 | 18.09 | 9.91 | 0.68 | 2.98 | 10.91 |
| | w/o align. | | 39.46 | 37.60 | 36.57 | 32.28 | 29.71 | 35.12 | 18.55 | 18.39 | 16.34 | 13.59 | 11.55 | 15.68 |
| | IMDC-DIA | | **52.03** | **46.47** | **44.22** | **41.54** | **39.14** | **44.68** | 27.04 | **25.11** | **21.99** | **20.89** | **17.73** | **22.55** |
| MSRCV1 | w/o imp. | avg | 72.86 | 59.52 | 52.22 | 44.60 | 38.10 | 53.46 | 52.89 | 27.30 | 20.49 | 10.75 | 7.80 | 23.85 |
| | | zero | 69.05 | 63.33 | 57.30 | 46.51 | 43.81 | 56.00 | 43.29 | 33.62 | 25.93 | 14.12 | 11.75 | 25.74 |
| | | rand | 67.46 | 56.83 | 51.59 | 46.03 | 44.92 | 53.37 | 48.75 | 31.53 | 26.17 | 15.75 | 12.57 | 26.96 |
| | | none | 79.68 | 62.38 | 62.22 | 33.97 | 41.43 | 55.94 | 66.16 | 46.72 | 39.52 | 8.04 | 14.72 | 35.03 |
| | w/o align. | | 85.56 | 85.24 | 73.65 | 67.62 | 62.86 | 74.98 | 72.51 | 70.51 | 52.78 | 43.35 | 39.98 | 55.83 |
| | IMDC-DIA | | **91.27** | **92.22** | **86.03** | **85.08** | **83.02** | **87.52** | **80.92** | **82.50** | **70.95** | **69.28** | **64.59** | **73.65** |

