# OpenReview forum: "Incomplete Multi-view Deep Clustering with Data Imputation and Alignment"
_NeurIPS.cc/2025/Conference — NeurIPS 2025 poster_

### Official Review · Reviewer_kzHN · 2025-06-20

**Clarity:** 2
**Significance:** 4
**Originality:** 4
**Rating:** 5
**Confidence:** 5

**Summary:**

Existing multi-view deep clustering methods encode the data observations into latent representations but ignore their uniqueness to all views. Also, they fail to consider the pair-wise similarities of missing data observations. To address the two problems, this paper proposes an incomplete multi-view deep clustering method with data imputation and alignment named IMDC-DIA.

**Questions:**

1. In the experiment setting, the authors claim that four metrics are adopted to evaluate the method. But there are only two of them, i.e. ACC and NMI, in the tables. So the authors are supposed to provide the results of PUR and ARI.

2. The missing data setting should be illustrated more clearly.

3. More details of experiment settings are expected to be provided, such as the neural network structure, learning rate, optimizer, etc.

4. The code should be released publicly for reproductivity.

**Ethical Concerns:**

["NO or VERY MINOR ethics concerns only"]

**Final Justification:**

Thanks to the author for solving my concerns, and I decided to keep the original score.

**Limitations:**

Yes, at the end of the Conclusion section.

**Quality:**

4

**Strengths And Weaknesses:**

Strength

1. Novelty. Different from the existing methods of encoding the data observation into latent representation, this paper assumes the latter is unique and projects it to reconstruct the former. This is rational and novel.

2. Writing. This paper is well-organized and easy to follow. Also, the framework of Fig. 1 provides a good overview of the method and is appreciated.

3. Effectiveness. The proposed method achieves the best performances over those in literature in almost all missing ratios. Also, the ablation study well validates effectiveness of the data imputation and alignment modules.


Weakness

1. Completeness of experiment. This paper lacks some experiment results.

2. Clarity of experiment setting. Some of the important experiment settings, such as details of deep learning structure, are missing.

- Please find the specifics in Questions section.

---

> ### Author Rebuttal · Authors · 2025-07-30
>
> Thanks so much for your recognition on the novelty, writing and effectiveness.
>
>
> **Q1. PUR and ARI results**
>
> Thanks. After running the experiments again, we have obtained the PUR and ARI results. In the following are those on Handwritten dataset, while the others will be provided in Appendix. It can be seen that the purity and ARI results follow the same trend with the ACC and NMI results.
>
> | Metric |  | PUR |  |  |  |  |  | ARI |  |  |  |  |  |
> |---|---|:---:|:---:|:---:|:---:|:---:|:---:|:---:|:---:|:---:|:---:|:---:|:---:|
> | Missing ratio |  | 0.1 | 0.3 | 0.5 | 0.7 | 0.9 | avg. | 0.1 | 0.3 | 0.5 | 0.7 | 0.9 | avg. |
> | HandWritten | DITA-IMVC | 77.85 | 80.90 | 81.38 | 81.02 | 55.00 | 75.23 | 66.61 | 70.78 | 70.87 | 68.67 | 35.92 | 62.57 |
> |  | DSIMVC | 81.13 | 80.73 | 79.88 | 77.50 | 51.18 | 74.08 | 71.69 | 70.19 | 67.27 | 63.91 | 32.36 | 61.08 |
> |  | DCP | 83.97 | 78.08 | 79.40 | 73.40 | 13.42 | 65.65 | 75.32 | 61.27 | 62.61 | 53.62 | 0.02 | 50.57 |
> |  | DVIMC | 86.53 | 83.12 | 45.53 | 25.27 | 19.45 | 51.98 | 82.38 | 77.91 | 35.65 | 14.20 | 5.74 | 43.18 |
> |  | CPSPAN | 90.42 | 91.25 | 91.08 | 90.27 | 87.82 | 90.17 | 80.83 | 82.12 | 81.69 | 80.46 | 78.02 | 80.62 |
> |  | IMDC-DIA | 96.37 | 93.68 | 91.80 | 90.65 | 87.93 | 92.09 | 92.09 | 93.68 | 91.80 | 90.65 | 87.93 | 91.23 |
>
>
> | Metric |  |  | PUR |  |  |  |  |  | ARI |  |  |  |  |  |
> |---|---|---|:---:|:---:|:---:|:---:|:---:|:---:|:---:|:---:|:---:|:---:|:---:|:---:|
> | Missing ratio |  |  | 0.1 | 0.3 | 0.5 | 0.7 | 0.9 | avg. | 0.1 | 0.3 | 0.5 | 0.7 | 0.9 | avg. |
> | HandWritten | w/o imp. | avg | 78.12 | 68.13 | 59.27 | 53.78 | 44.48 | 60.76 | 62.70 | 44.29 | 24.49 | 18.14 | 11.49 | 32.22 |
> |  |  | zero | 83.98 | 70.98 | 62.73 | 51.35 | 46.63 | 63.14 | 71.39 | 47.75 | 32.75 | 21.58 | 15.48 | 37.79 |
> |  |  | rand | 80.18 | 71.13 | 59.95 | 55.62 | 46.58 | 62.69 | 70.79 | 55.61 | 32.89 | 25.63 | 15.86 | 40.16 |
> |  |  | none | 96.13 | 84.73 | 89.28 | 79.67 | 56.68 | 81.30 | 91.59 | 77.60 | 80.11 | 69.16 | 37.18 | 71.13 |
> |  | w/o align. |  | 93.62 | 88.18 | 90.57 | 83.77 | 84.12 | 88.05 | 86.73 | 78.23 | 80.55 | 71.71 | 69.92 | 77.43 |
> |  | IMDC-DIA |  | 96.37 | 93.68 | 91.80 | 90.65 | 87.93 | 92.09 | 92.09 | 86.37 | 82.60 | 80.53 | 75.42 | 83.40 |
>
>
> **Q2. Missing data setting**
>
> Thanks. We generate incomplete datasets by following the common strategy [36] in literature.
> Specifically, assuming the missing ratio to be $m$, $m$ percent of data samples are selected to remove at least one views randomly.
> In the experiments, $m$ is set in $\\{0.1, 0.3, 0.5, 0.7. 0.9\\}$.
> Nevertheless, we provide the corresponding pseudocode and will add it in Appendix.
>
> **Algorithm 2. Generation of Incomplete Multi-view Data**
> **Input**: incomplete multi-view data $\\{\mathbf{x}_ i^{(v)}\\}_ {i,v=1}^{N,V}$, missing ratio $m$
> **Output**: availability indicator $\\{\mathbf{a}_ i\\}_ {i=1}^N$
> 1. obtain the numbers of data samples and views, i.e. $N$ and $V$;
> 2. random sample incomplete data index $\mathcal{S}$ with missing ratio $m$;
> 3. initialize availability indicator $\mathbf{a}_i = \\{1, 2, \cdots, V\\}$ w.r.t. $i\in \\{1, 2, \cdots, N\\}$;
> 4. **for** $p\in\mathcal{S}$ **do**
>     - random sample missing index $\mathbf{m}_p \subset \\{1, 2, \cdots, V\\}$ and $\mathbf{m}_p = \emptyset$;
>     - set availability indicator $\mathbf{a}_p = \\{1, 2, \cdots, V\\} - \mathbf{m}_p$;
> 5. **end for**
>
>
> **Q3. More experiment setting**
>
> Thanks. We provide the details of experiment setting in the following and will add them in Appendix.
>
> In the experiments, we adopt fully-connected neural networks on all datasets where different numbers of neurons are adopted according to different feature dimensions, as shown in the following table.
> Meanwhile, the unique data latent representation and parameters of neural networks are both optimized with gradient descent strategy with Adam optimizer whose learning rate is set to 0.001.
> Additionally, the codes of the proposed method and recent advances in comparison are executed on a server with 40 *Intel(R) Xeon(R) Silver 4210R CPU @ 2.40GHz* CPUs and 2 *NVIDIA GeForce RTX 3090* GPUs.
> The software environment includes *Python 3.10.13* and *PyTorch 1.12.1* optimized with *CUDA 11.4*.
>
> |  | HandWritten | Caltech5V | Flower17 | MSRCV1 |
> |---|---|---|---|---|
> | 1st-view | (60, 64, 216) | (21, 16, 40) | (34, 512, 5376) | (21, 256, 1302) |
> | 2nd-view | (60,64,76) | (21, 64, 254) | (34, 256, 512) | (21, 32, 48) |
> | 3rd-view | (60, 32, 64) | (21, 128, 928) | (34, 512, 5376) | (21, 64, 512) |
> | 4th-view  | (60, 16, 6) | (21, 128, 512) | (34, 512, 5376) | (21, 32, 100) |
> | 5th-view | (60, 64, 240) | (21, 256, 1984) | (34, 256, 1239) | (21, 64, 256) |
> | 6th-view | (40, 32, 47) | - | (34, 512, 5376) | (21, 64, 210) |
> | 7th-view | - | - | (34, 512, 5376) | - |
>
>
> **Q4. Code release**
>
> Thanks. The code will be released once the paper is accepted.

---

> > ### Comment · Reviewer_kzHN · 2025-08-05
> >
> > Thanks to the authors for the response. I have a few additional minor comments:
> >
> > In addition to the problem of Equation (1), the expression in line 148 — “Instead of encoding the available data observations into multiple view-specific latent representations...” , seems slightly contradictory to the subsequent statement that the latent representations “should be unique to all data views.”
> >
> > It is recommended to annotate $\bar{x}$  and $\hat{x}$ in the framework diagram, as this would improve clarity.

---

> > > ### Author Response · Authors · 2025-08-06
> > > **The Response**
> > >
> > > Thanks for your recognition and feedback!
> > >
> > > **Q1. Slightly contradictory expression**
> > >
> > > Thanks. Here, we are to say that the existing methods encode the available data observations into multiple view-specific latent representations where each data view corresponds to one latent representation. But this is contradictory to the common sense that  the latent representation only depends on the data intrinsic structure and therefore should be unique to all data views. On the basis, the proposed IMDC-DIA method assumes all data views share a same latent representation. So the expression is correct but vague. We sincerely apologize for this and will modify it more clearly in next version.
> > >
> > > **Q2. Clarity of framework diagram**
> > >
> > > Thanks. We really appreciate the advice and will add the annotation $\bar{x}$ and $\hat{x}$ in the framework diagram to improve clarity.

---

> > > > ### Comment · Reviewer_kzHN · 2025-08-09
> > > >
> > > > Thanks to the author for solving my concerns, and I decided to keep the original score.

---

### Official Review · Reviewer_ZBHB · 2025-06-24

**Clarity:** 3
**Significance:** 3
**Originality:** 3
**Rating:** 4
**Confidence:** 5

**Summary:**

To consider the uniqueness of data representation of all data views and the pair-wise similarities of missing data observations, the authors propose an incomplete multi-view deep clustering method by imputing the missing data and aligning them iteratively. Moreover, they conduct experiments on the method and achieve state-of-the-art performances.

**Questions:**

Most issues have already been mentioned in Weaknesses. One additional problem is as follows:
- In Theorem 1, the authors say "The linear alignment of two arbitrary matrices measures the consistency of the pair-wise linear similarities of their rows," but they do not define what "consistency" means or "pair-wise linear similarities". The authors should clearly define "pair-wise linear similarities" and "consistency" in Theorem 1 and Proof 1.

**Ethical Concerns:**

["NO or VERY MINOR ethics concerns only"]

**Final Justification:**

Thank the author for their rebuttal. My concerns have been resolved. I will raise my score to 'Borderline accept'.

**Limitations:**

Yes.

**Paper Formatting Concerns:**

No concerns.

**Quality:**

3

**Strengths And Weaknesses:**

Strengths

- The authors propose a new measurement (linear alignment) similar to kernel alignment which reduces the computation complexity to O(n). Also, the linear alignment is validated to be effective in experiments, especially in ablation study of Table 2. This will have a broader impact on the future research.

- The experiment design of ablation study is impressive. Insides, the w/o imp. and w/o align. settings well validate the functions of data imputation and data alignment components respectively. Also, the w/o imp. setting considers four strategies, i.e. the average-filling, zero-filling, rand-filling and no-filling, which is complete.

- The IDMC-DIA method achieves state-of-the-art results compared with the existing methods in literature, well validating its effectiveness.

- The limitations are discussed. Also, the possible future work is also provided, which is appreciated.

Weaknesses

- As seen from Fig. 1 and corresponding descriptions, the $z_i^{(v)}$ in Equation (1) is supposed to be $z_i$ probably. The authors should check this.

- In Equation (3), $v$ lacks the bracket. Please keep consistent in the whole paper.

- In optimization section, the authors provide the method to optimize the neural network parameter, latent representation and weight parameter. But how to combine them in order is not illustrated clearly. The authors are encouraged to provide the pseudocode.

- The convergence should be analyzed.

- Table 1 and 2 only present the ACC and NMI results, but miss the PUR and ARI. Although the paper length is limited, the authors may provide them in Appendix.

---

> ### Author Rebuttal · Authors · 2025-07-30
>
> Thanks so much for your recognition on the proposed alignment measurement, experiment design and effectiveness!
>
> **Q1 and Q2. Typos**
>
> Thanks. So sorry for the typos! $z_i^{(v)}$ should be revised to $z_i$, where Eq. (1) should be
> $\hat{\mathbf{x}}_i^{(v)} = g _{\Theta_v}(\mathbf{z}_i)$
> Also, we have added the bracket of $v$ in Eq. (3) to keep consistency of all equations, i.e.
> $$\bar{\mathbf{x}}_i^{(v)} = \begin{cases}
>     \mathbf{x}_i^{(v)} & \text{ if } \mathbf{a} _{i,v} = 1, \\\\
>     \hat{\mathbf{x}}_i^{(v)} & \text{ if } \mathbf{a} _{i,v} = 0.
>   \end{cases}$$
> Nevertheless, we have proofread the whole manuscript to eliminate all typos and inconsistencies.
>
> **Q3. Algorithm pseudocode**
>
> Thanks. We have re-organized the optimization section and added the corresponding pseudocode, which are provided in the following.
>
> According to the overall loss of Eq. (9), there are three variables to optimize in model training, including the neural network parameter $\\{\Theta_v\\}_ {v=1}^V$, the unique latent representation $\\{\mathbf{z}_ i\\}_ {i=1}^N$ and weight parameter $\\{w_v\\}_ {v=1}^V$.
> Each of them can be optimized as follows:
> - *Optimization of neural network parameter $\\{\Theta_v\\}_ {v=1}^V$*. Same to most of deep learning methods, the neural network parameter can be optimized with gradient descent strategy. In experiment, we adopt the popular Adaptive Moment Estimation (Adam) optimizer.
> - *Optimization of latent representation $\\{\mathbf{z}_ i\\}_ {i=1}^N$*. Different from the existing multi-view clustering methods, such as [10], the proposed IMDC-DIA method is difficult to find the close-form solution of data latent representation. Therefore, the gradient descent strategy with Adam optimizer is adopted in its optimization.
> - *Optimization of weight parameter $\\{w_v\\}_ {v=1}^V$*. With fixing the others, $L_a^{(v)}$ is given and minimizing the overall loss of Eq. (9) equals to
> $$
>   \max_ {\\{w_v\\}_ {v=1}^V} \sum_ {v=1}^V w_v(-L_a^{(v)}), \quad s.t. \sum_ {v=1}^V w_v^2 = 1.
> $$
> According to o Cauchy–Schwarz inequality [31],
> $$
>   \sum_{v=1}^V w_v(-L_a^{(v)}) \leq \sqrt{\left(\sum_{v=1}^Vw_v^2\right)\left(\sum_{v=1}^VL_a^{(v)2}\right)} = \sqrt{\sum_{v=1}^VL_a^{(v)2}},
> $$
> where the equality holds for when
> $$
>   \frac{w_1}{-L_a^{(1)}} = \frac{w_2}{-L_a^{(2)}} = \cdots = \frac{w_V}{-L_a^{(V)}}.
> $$
> Unifying the constraint $\sum_ {v=1}^V w_v^2 = 1$ in Eq. (10), the solution can be computed to
> $$
>   w_v^* = -L_a^{(v)} / \sqrt{\sum_{v'=1}^V L_a^{(v')2}}.
> $$
> Overall, the aforementioned three parameters are optimized alternately and the latent representations can be achieved until the convergence or maximal epoch.
> Next, $k$-means algorithm is applied on the computed latent representations $\\{\mathbf{z}_i\\} _ {i=1}^N$ to group multi-view data into categories.
> Furthermore, to specify the optimization procedure more clearly, corresponding pseudo-code is summarized in Alg. 1.
>
>
> **Algorithm 1. Incomplete Multi-view Deep Clustering with Data Imputation and Alignment**
> **Input**: incomplete multi-view data $\\{\mathbf{x}_ i^{(v)}\\}_ {i,v=1}^{N,V}$ with availability indicator $\\{\mathbf{a}_ i\\}_ {i=1}^N$, cluster number $k$
> **Output**: data labels
> 1. initialize latent representation $\\{\mathbf{z}_ i\\}_ {i=1}^N$ and network parameters $\\{\Theta_ v\\}_ {v=1}^V$;
> 2. $t=0$;
> 3. **while** $t<epochs$
>     - *\# forward*
>     - compute $L_r$ with Eq. (2);
>     - complete multi-view data with Eq. (3);
>     - compute $\\{L_a^{(v)}\\}_ {v=1}^V$ with Eq. (7);
>     - update the view weights $\\{w_v\\}_ {v=1}^V$ with Eq. (13);
>     - compute the overall loss $L$ with Eq. (9);
>     - *\# back propagation*
>     - update latent representation $\\{\mathbf{z}_ i\\}_ {i=1}^{N}$ and network parameters $\\{\Theta_v\\}_ {v=1}^V$;
>     - $t = t + 1$;
> 4. **end while**
> 5. compute the data labels with $k$-means on latent representations $\\{\mathbf{z}_ i\\}_ {i=1}^N$;
>
>
> **Q4. Convergence analysis**
>
> Thanks. we analyze the convergence of the proposed algorithm as follows.
>
> In the optimization, the proposed IMDC-DIA method aims to minimize the loss $L$ of Eq. (9).
> Insides, three variables are optimized alternately, including the neural network parameter $\\{\Theta_v\\}_ {v=1}^V$, the unique latent representation $\\{\mathbf{z}_ i\\}_ {i=1}^N$ and weight parameter $\\{w_v\\}_ {v=1}^V$.
> With a little abuse of notation, let the loss be represented by
> \begin{equation}
>   L = \ell(\\{\Theta_ v\\}_ {v=1}^V, \\{\mathbf{z}_ i\\}_ {i=1}^N, \\{w_v\\}_ {v=1}^V).
> \end{equation}
> Taking the loss at $t$ epoch to be
> \begin{equation}
>   L^{(t)} = \ell(\\{\Theta_ v^{(t)}\\}_ {v=1}^V, \\{\mathbf{z}^{(t)}_ i\\}_ {i=1}^N, \\{w_ v^{(t)}\\}_ {v=1}^V),
> \end{equation}
> By fixing $\\{\mathbf{z}_ i\\}_ {i=1}^N$ and $\\{w_v\\}_ {v=1}^V$ at $\\{\mathbf{z}^{(t)}_ i\\}_ {i=1}^N$ and $\\{w_ v^{(t)}\\}_ {v=1}^V$, optimizing $\\{\Theta_ v\\}_ {v=1}^V$ at $t+1$ epoch induces to
> \begin{equation}
>   \ell(\\{\Theta_ v^{(t)}\\}_ {v=1}^V, \\{\mathbf{z}^{(t)}_ i\\}_ {i=1}^N, \\{w_ v^{(t)}\\}_ {v=1}^V) \leq \ell(\\{\Theta_ v^{(t+1)}\\}_ {v=1}^V, \\{\mathbf{z}^{(t)}_ i\\}_ {i=1}^N, \\{w_ v^{(t)}\\}_ {v=1}^V).
> \end{equation}
> So do the optimizations of $\\{\mathbf{z}_ i\\}_ {i=1}^N$ and $\\{w_ v\\}_ {v=1}^V$, resulting in
> \begin{equation}
> \begin{split}
>   & \ell(\\{\Theta_ v^{(t)}\\}_ {v=1}^V, \\{\mathbf{z}^{(t)}_ i\\}_ {i=1}^N, \\{w_ v^{(t)}\\}_ {v=1}^V) \leq \ell(\\{\Theta_ v^{(t+1)}\\}_ {v=1}^V, \\{\mathbf{z}^{(t)}_ i\\}_ {i=1}^N, \\{w_ v^{(t)}\\}_ {v=1}^V) \\\\
>   & \leq \ell(\\{\Theta_ v^{(t+1)}\\}_ {v=1}^V, \\{\mathbf{z}^{(t+1)}_ i\\}_ {i=1}^N, \\{w_ v^{(t)}\\}_ {v=1}^V) \leq \ell(\\{\Theta_ v^{(t+1)}\\}_ {v=1}^V, \\{\mathbf{z}^{(t+1)}_ i\\}_ {i=1}^N, \\{w_ v^{(t+1)}\\}_ {v=1}^V),
> \end{split}
> \end{equation}
> which can be simplified to
> \begin{equation}
>   L^{(t)} \leq L^{(t+1)}.
> \end{equation}
> Nevertheless,
> \begin{equation}
>   L = L_r + \beta L_a \geq 0 + (-1) = -1.
> \end{equation}
> To be summarized, the former equation indicates that the loss monotonically decreases, while the latter illustrates that the loss is lower bounded.
> Therefore, the optimization algorithm is convergent.
>
> **Q5. PUR and ARI results**
>
> Thanks. After running the experiments again, we have obtained the PUR and ARI results. In the following are those on Handwritten dataset, while the others will be provided in Appendix. It can be seen that the purity and ARI results follow the same trend with the ACC and NMI results.
>
> | Metric |  | PUR |  |  |  |  |  | ARI |  |  |  |  |  |
> |---|---|:---:|:---:|:---:|:---:|:---:|:---:|:---:|:---:|:---:|:---:|:---:|:---:|
> | Missing ratio |  | 0.1 | 0.3 | 0.5 | 0.7 | 0.9 | avg. | 0.1 | 0.3 | 0.5 | 0.7 | 0.9 | avg. |
> | HandWritten | DITA-IMVC | 77.85 | 80.90 | 81.38 | 81.02 | 55.00 | 75.23 | 66.61 | 70.78 | 70.87 | 68.67 | 35.92 | 62.57 |
> |  | DSIMVC | 81.13 | 80.73 | 79.88 | 77.50 | 51.18 | 74.08 | 71.69 | 70.19 | 67.27 | 63.91 | 32.36 | 61.08 |
> |  | DCP | 83.97 | 78.08 | 79.40 | 73.40 | 13.42 | 65.65 | 75.32 | 61.27 | 62.61 | 53.62 | 0.02 | 50.57 |
> |  | DVIMC | 86.53 | 83.12 | 45.53 | 25.27 | 19.45 | 51.98 | 82.38 | 77.91 | 35.65 | 14.20 | 5.74 | 43.18 |
> |  | CPSPAN | 90.42 | 91.25 | 91.08 | 90.27 | 87.82 | 90.17 | 80.83 | 82.12 | 81.69 | 80.46 | 78.02 | 80.62 |
> |  | IMDC-DIA | 96.37 | 93.68 | 91.80 | 90.65 | 87.93 | 92.09 | 92.09 | 93.68 | 91.80 | 90.65 | 87.93 | 91.23 |
>
>
> | Metric |  |  | PUR |  |  |  |  |  | ARI |  |  |  |  |  |
> |---|---|---|:---:|:---:|:---:|:---:|:---:|:---:|:---:|:---:|:---:|:---:|:---:|:---:|
> | Missing ratio |  |  | 0.1 | 0.3 | 0.5 | 0.7 | 0.9 | avg. | 0.1 | 0.3 | 0.5 | 0.7 | 0.9 | avg. |
> | HandWritten | w/o imp. | avg | 78.12 | 68.13 | 59.27 | 53.78 | 44.48 | 60.76 | 62.70 | 44.29 | 24.49 | 18.14 | 11.49 | 32.22 |
> |  |  | zero | 83.98 | 70.98 | 62.73 | 51.35 | 46.63 | 63.14 | 71.39 | 47.75 | 32.75 | 21.58 | 15.48 | 37.79 |
> |  |  | rand | 80.18 | 71.13 | 59.95 | 55.62 | 46.58 | 62.69 | 70.79 | 55.61 | 32.89 | 25.63 | 15.86 | 40.16 |
> |  |  | none | 96.13 | 84.73 | 89.28 | 79.67 | 56.68 | 81.30 | 91.59 | 77.60 | 80.11 | 69.16 | 37.18 | 71.13 |
> |  | w/o align. |  | 93.62 | 88.18 | 90.57 | 83.77 | 84.12 | 88.05 | 86.73 | 78.23 | 80.55 | 71.71 | 69.92 | 77.43 |
> |  | IMDC-DIA |  | 96.37 | 93.68 | 91.80 | 90.65 | 87.93 | 92.09 | 92.09 | 86.37 | 82.60 | 80.53 | 75.42 | 83.40 |
>
>
> **Q6. Clarity of Theorem 1**
>
> Thanks. Sorry for the missing of these two concepts! They are introduced in detail. Also, we polish Theorem 1 and its proof carefully. Due to the space limit, please refer to the **Q6** of **Reviewer aouo** (*the same question*).

---

> > ### Comment · Reviewer_ZBHB · 2025-08-06
> >
> > Thanks for the rebuttal. They have addressed my concerns and I decide to raise my score.

---

> > > ### Author Response · Authors · 2025-08-06
> > > **The Response**
> > >
> > > Thanks so much for your recognition!

---

> ### Author Response · Authors · 2025-08-05
> **The Kind Rebuttal Inquiry**
>
> Dear Reviewer ZBHB,
>
> Thanks so much for your valuable comments! They help us a lot to improve the quality of this paper. As the discussion phase deadline is approaching, could you please check the responses? If you have any further questions, we are glad to answer.
>
> Best regards,
> The authors

---

### Official Review · Reviewer_Awrn · 2025-06-26

**Clarity:** 3
**Significance:** 3
**Originality:** 3
**Rating:** 5
**Confidence:** 5

**Summary:**

This paper proposes an incomplete multi-view deep clustering method with data imputation and alignment named IMDC-DIA. It assumes that the latent representations are unique to a fixed set of data samples in all views. Moreover, it utilizes the pair-wise similarities of missing data observations in latent representation learning. In experiments, the authors compare it with the existing methods on four datasets. It can be seen that the IMDC-DIA method obtains state-of-the-art performances.

**Questions:**

See strengths and weaknesses.

**Ethical Concerns:**

["NO or VERY MINOR ethics concerns only"]

**Final Justification:**

The authors' response have addressed most of my concerns, thus I will raise my score.

**Limitations:**

yes

**Paper Formatting Concerns:**

This paper follows the formatting instructions.

**Quality:**

3

**Strengths And Weaknesses:**

**Strengths**

1. The proposed method assumes that the latent representations are unique to a fixed set of data samples in all views, which is different from that of the most existing methods. This is rational and more consistent to real-world practice.

2. The proposed linear alignment considers but does not require to compute the pair-wise similarities of data samples between different views, hence is of linear complexity to sample number. This is novel and practical, especially in large-scale problem.

3. The proposed method is only composed of three simple and direct modules, i.e. data reconstruction, imputation and alignment. No sophisticated techniques are derivations are involved. But it achieves impressive performances in the comparison experiments. So the method is elegant and effective.

4. The paper is well-organized and the writing is easy to follow.

**Weaknesses**

1. In Eq. (9), the trade-off parameter is $\alpha$, but the parameter is written to $\beta$ mistakenly in Fig. 2 and corresponding analysis. Please keep them consistent.

2. In experiment, the performance and loss to training epoch are supposed to be presented and analyzed, at least for validating the rationality of the loss design.

3. It is claimed that PUR and ARI are adopted in experiments in Section 4.1. However, they are missing in Table 1 and 2.

4. Fig. 2 presents the accuracy variation to parameter $\beta$. Also, the other evaluation measurements should be provided, including NMI, PUR and ARI.

5. The optimization should be enhanced. At least, the optimization order and logic should be clarified.

6. Please eliminate all typos and grammar errors.

7. In Theorem 1, the authors say "The linear alignment of two arbitrary matrices measures the consistency of the pair-wise linear similarities of their rows," but they do not define what "consistency" means or "pair-wise linear similarities." So, Theorem 1 does not make sense at all in the current manuscript.

---

> ### Author Rebuttal · Authors · 2025-07-30
>
> Thanks so much for your recognition on the novelty, rationality, effectiveness and paper writing!
>
> **Q1. Typos**
>
> Thanks. Sorry for the typos! we have replaced $\alpha$ with $\beta$ in the whole manuscript.
>
>
> **Q2. Performance and loss variation in training process**
>
> Thanks.
> We have recorded the performance and loss value at each training epoch. Since the figures cannot be uploaded in Rebuttal period, we list the accuracy values on HandWritten dataset at each 40 epochs in the following:
>
> `
> Accuracy: 12.80, 18.60, 50.50, 75.10, 85.90, 88.70, 89.75, 89.10, 79.00, 90.30, 92.15, 92.90, 92.70, 92.10, 94.20, 95.00, 94.95, 94.60, 84.80, 93.85, 93.45, 95.60, 95.70, 93.95, 95.95, 95.90, 95.85, 94.30, 82.20, 96.40, 96.35, 96.35, 96.20, 96.55, 83.45, 96.50, 96.50, 96.25, 96.65, 96.70, 96.65, 96.40, 96.75, 96.55, 97.00, 96.85, 96.80, 82.80, 96.90, 96.65
> `
>
> `
> Loss value: 1.4718, 0.4360, 0.1024, 0.0549, 0.0281, -0.0070, -0.0433, -0.0717, -0.0925, -0.1082, -0.1203, -0.1298, -0.1375, -0.1437, -0.1489, -0.1534, -0.1573, -0.1607, -0.1637, -0.1663, -0.1687, -0.1708, -0.1726, -0.1743, -0.1757, -0.1769, -0.1780, -0.1790, -0.1799, -0.1807, -0.1814, -0.1821, -0.1828, -0.1834, -0.1839, -0.1845, -0.1850, -0.1854, -0.1859, -0.1863, -0.1867, -0.1870, -0.1874, -0.1877, -0.1880, -0.1883, -0.1886, -0.1888, -0.1891, -0.1893
> `
>
> It can be observed that the loss value continuously decreases and finally converges to the minimum, while the accuracy increases to the top and fluctuate around their top values.
> These observations well illustrate the effectiveness of loss design in Eq. (9) and the proposed IMDC-DIA method.
>
> In next version, we will plot them and provide the loss value variation on all datasets in Appendix.
>
>
> **Q3. Missing PUR and ARI results**
>
> Thanks. After running the experiments again, we have obtained the PUR and ARI results. In the following are those on Handwritten dataset, while the others will be provided in Appendix. It can be seen that the purity and ARI results follow the same trend with the ACC and NMI results.
>
> | Metric |  | PUR |  |  |  |  |  | ARI |  |  |  |  |  |
> |---|---|:---:|:---:|:---:|:---:|:---:|:---:|:---:|:---:|:---:|:---:|:---:|:---:|
> | Missing ratio |  | 0.1 | 0.3 | 0.5 | 0.7 | 0.9 | avg. | 0.1 | 0.3 | 0.5 | 0.7 | 0.9 | avg. |
> | HandWritten | DITA-IMVC | 77.85 | 80.90 | 81.38 | 81.02 | 55.00 | 75.23 | 66.61 | 70.78 | 70.87 | 68.67 | 35.92 | 62.57 |
> |  | DSIMVC | 81.13 | 80.73 | 79.88 | 77.50 | 51.18 | 74.08 | 71.69 | 70.19 | 67.27 | 63.91 | 32.36 | 61.08 |
> |  | DCP | 83.97 | 78.08 | 79.40 | 73.40 | 13.42 | 65.65 | 75.32 | 61.27 | 62.61 | 53.62 | 0.02 | 50.57 |
> |  | DVIMC | 86.53 | 83.12 | 45.53 | 25.27 | 19.45 | 51.98 | 82.38 | 77.91 | 35.65 | 14.20 | 5.74 | 43.18 |
> |  | CPSPAN | 90.42 | 91.25 | 91.08 | 90.27 | 87.82 | 90.17 | 80.83 | 82.12 | 81.69 | 80.46 | 78.02 | 80.62 |
> |  | IMDC-DIA | 96.37 | 93.68 | 91.80 | 90.65 | 87.93 | 92.09 | 92.09 | 93.68 | 91.80 | 90.65 | 87.93 | 91.23 |
>
>
> | Metric |  |  | PUR |  |  |  |  |  | ARI |  |  |  |  |  |
> |---|---|---|:---:|:---:|:---:|:---:|:---:|:---:|:---:|:---:|:---:|:---:|:---:|:---:|
> | Missing ratio |  |  | 0.1 | 0.3 | 0.5 | 0.7 | 0.9 | avg. | 0.1 | 0.3 | 0.5 | 0.7 | 0.9 | avg. |
> | HandWritten | w/o imp. | avg | 78.12 | 68.13 | 59.27 | 53.78 | 44.48 | 60.76 | 62.70 | 44.29 | 24.49 | 18.14 | 11.49 | 32.22 |
> |  |  | zero | 83.98 | 70.98 | 62.73 | 51.35 | 46.63 | 63.14 | 71.39 | 47.75 | 32.75 | 21.58 | 15.48 | 37.79 |
> |  |  | rand | 80.18 | 71.13 | 59.95 | 55.62 | 46.58 | 62.69 | 70.79 | 55.61 | 32.89 | 25.63 | 15.86 | 40.16 |
> |  |  | none | 96.13 | 84.73 | 89.28 | 79.67 | 56.68 | 81.30 | 91.59 | 77.60 | 80.11 | 69.16 | 37.18 | 71.13 |
> |  | w/o align. |  | 93.62 | 88.18 | 90.57 | 83.77 | 84.12 | 88.05 | 86.73 | 78.23 | 80.55 | 71.71 | 69.92 | 77.43 |
> |  | IMDC-DIA |  | 96.37 | 93.68 | 91.80 | 90.65 | 87.93 | 92.09 | 92.09 | 86.37 | 82.60 | 80.53 | 75.42 | 83.40 |
>
>
>
> **Q4. NMI, PUR and ARI results on parameter study**
>
> Thanks. After running the experiments again, we have computed the NMI, PUR and ARI results when parameter $\beta$ varies in range $\\{0.01, 0.1, 1, 10, 100\\}$. Since the figures cannot be uploaded in Rebuttal period, we list them in the following table.
>
> - The NMI results with respect to different $\beta$ in different missing ratios.
> | $\beta$ | 0.1 | 0.3 | 0.5 | 0.7 | 0.9 |
> |---|---|---|---|---|---|
> | 0.01 | 90.54 | 85.32 | 83.31 | 82.05 | 76.77 |
> | 0.1 | 91.80 | 86.68 | 83.67 | 81.05 | 77.39 |
> | 1 | 86.99 | 77.55 | 72.86 | 67.22 | 66.71 |
> | 10 | 83.42 | 75.21 | 68.92 | 61.40 | 57.90 |
> | 100 | 81.64 | 71.65 | 62.97 | 55.85 | 43.97 |
>
> - The PUR results with respect to different $\beta$ in different missing ratios.
> | $\beta$ | 0.1 | 0.3 | 0.5 | 0.7 | 0.9 |
> |---|---|---|---|---|---|
> | 0.01 | 95.65 | 90.62 | 91.80 | 90.65 | 85.23 |
> | 0.1 | 96.37 | 93.68 | 89.38 | 90.37 | 87.93 |
> | 1 | 88.72 | 80.08 | 76.95 | 71.30 | 72.72 |
> | 10 | 84.58 | 77.75 | 73.57 | 67.98 | 64.47 |
> | 100 | 83.55 | 76.48 | 69.17 | 62.22 | 49.47 |
>
> - The ARI results with respect to different $\beta$ in different missing ratios.
> | $\beta$ | 0.1 | 0.3 | 0.5 | 0.7 | 0.9 |
> |---|---|---|---|---|---|
> | 0.01 | 90.61 | 83.15 | 82.60 | 80.53 | 72.80 |
> | 0.1 | 92.09 | 86.37 | 81.86 | 79.84 | 75.42 |
> | 1 | 83.70 | 71.08 | 63.92 | 54.18 | 53.81 |
> | 10 | 78.90 | 66.68 | 55.82 | 44.31 | 37.59 |
> | 100 | 76.86 | 63.46 | 48.54 | 37.57 | 24.23 |
>
> It can be observed that those metrics follows similar trend to the ACC. Specifically, they decrease with a larger $\beta$ in almost all missing ratios on all benchmark datasets, while the best are obtained mostly when setting $\beta$ to 0.01 and sometimes when setting $\beta$ to 0.1. Therefore, we recommend to set parameter $\beta$ from 0.01 to 0.1 in first priority.
>
>
> **Q5. Clarity of optimization**
>
> Thanks. We re-organize the optimization section and add the optimization pseudo-code which will be provided in Appendix. Meanwhile, they are provided in the following.
>
> According to the overall loss of Eq. (9), there are three variables to optimize in model training, including the neural network parameter $\\{\Theta_v\\}_ {v=1}^V$, the unique latent representation $\\{\mathbf{z}_ i\\}_ {i=1}^N$ and weight parameter $\\{w_v\\}_ {v=1}^V$.
> Each of them can be optimized as follows:
> - *Optimization of neural network parameter $\\{\Theta_v\\}_ {v=1}^V$*. Same to most of deep learning methods, the neural network parameter can be optimized with gradient descent strategy. In experiment, we adopt the popular Adaptive Moment Estimation (Adam) optimizer.
> - *Optimization of latent representation $\\{\mathbf{z}_ i\\}_ {i=1}^N$*. Different from the existing multi-view clustering methods, such as [10], the proposed IMDC-DIA method is difficult to find the close-form solution of data latent representation. Therefore, the gradient descent strategy with Adam optimizer is adopted in its optimization.
> - *Optimization of weight parameter $\\{w_v\\}_ {v=1}^V$*. With fixing the others, $L_a^{(v)}$ is given and minimizing the overall loss of Eq. (9) equals to
> $$
>   \max_ {\\{w_v\\}_ {v=1}^V} \sum_ {v=1}^V w_v(-L_a^{(v)}), \quad s.t. \sum_ {v=1}^V w_v^2 = 1.
> $$
> According to o Cauchy–Schwarz inequality [31],
> $$
>   \sum_{v=1}^V w_v(-L_a^{(v)}) \leq \sqrt{\left(\sum_{v=1}^Vw_v^2\right)\left(\sum_{v=1}^VL_a^{(v)2}\right)} = \sqrt{\sum_{v=1}^VL_a^{(v)2}},
> $$
> where the equality holds for when
> $$
>   \frac{w_1}{-L_a^{(1)}} = \frac{w_2}{-L_a^{(2)}} = \cdots = \frac{w_V}{-L_a^{(V)}}.
> $$
> Unifying the constraint $\sum_ {v=1}^V w_v^2 = 1$ in Eq. (10), the solution can be computed to
> $$
>   w_v^* = -L_a^{(v)} / \sqrt{\sum_{v'=1}^V L_a^{(v')2}}.
> $$
> Overall, the aforementioned three parameters are optimized alternately and the latent representations can be achieved until the convergence or maximal epoch.
> Next, $k$-means algorithm is applied on the computed latent representations $\\{\mathbf{z}_i\\} _ {i=1}^N$ to group multi-view data into categories.
> Furthermore, to specify the optimization procedure more clearly, corresponding pseudo-code is summarized in Alg. 1.
>
>
> **Algorithm 1. Incomplete Multi-view Deep Clustering with Data Imputation and Alignment**
> **Input**: incomplete multi-view data $\\{\mathbf{x}_ i^{(v)}\\}_ {i,v=1}^{N,V}$ with availability indicator $\\{\mathbf{a}_ i\\}_ {i=1}^N$, cluster number $k$
> **Output**: data labels
> 1. initialize latent representation $\\{\mathbf{z}_ i\\}_ {i=1}^N$ and network parameters $\\{\Theta_ v\\}_ {v=1}^V$;
> 2. $t=0$;
> 3. **while** $t<epochs$
>     - *\# forward*
>     - compute $L_r$ with Eq. (2);
>     - complete multi-view data with Eq. (3);
>     - compute $\\{L_a^{(v)}\\}_ {v=1}^V$ with Eq. (7);
>     - update the view weights $\\{w_v\\}_ {v=1}^V$ with Eq. (13);
>     - compute the overall loss $L$ with Eq. (9);
>     - *\# back propagation*
>     - update latent representation $\\{\mathbf{z}_ i\\}_ {i=1}^{N}$ and network parameters $\\{\Theta_v\\}_ {v=1}^V$;
>     - $t = t + 1$;
> 4. **end while**
> 5. compute the data labels with $k$-means on latent representations $\\{\mathbf{z}_ i\\}_ {i=1}^N$;
>
>
> **Q6. Typos and grammar errors**
>
> Thanks. We have carefully proofread the whole manuscript and tried our best to eliminate all typos and grammar errors. For example, the bracket is added on $v$ of Eq. (3) to keep consistency of all equations. More are not listed.
>
> **Q7. Clarity of Theorem 1**
>
> Thanks. Sorry for the missing of these two concepts! They are introduced in detail. Also, we polish Theorem 1 and its proof carefully. Due to the space limit, please refer to the **Q6** of **Reviewer aouo** (*the same question*).

---

### Official Review · Reviewer_aouo · 2025-06-27

**Clarity:** 3
**Significance:** 3
**Originality:** 4
**Rating:** 5
**Confidence:** 5

**Summary:**

Existing incomplete multi-view deep clustering methods typically encode each data view into an independent latent representation, overlooking the intrinsic uniqueness of the underlying data representation across views. Moreover, they fail to sufficiently exploit the pair-wise similarities among missing data instances. To address these limitations, this paper proposes a novel incomplete multi-view deep clustering approach that jointly imputes missing data and aligns the reconstructed views using a proposed linear alignment mechanism. The method enforces consistency across views by assuming a shared latent representation for each sample and projects it to each view via independent neural networks. Extensive experiments on four benchmark datasets demonstrate that the proposed approach consistently outperforms state-of-the-art methods across a wide range of missing ratios, validating its effectiveness in both representation learning and clustering accuracy.

**Questions:**

Please find in *Strengths and Weaknesses* section.

**Ethical Concerns:**

["NO or VERY MINOR ethics concerns only"]

**Final Justification:**

I thank the authors for their detailed rebuttal. I believe they have adequately addressed the main concerns I raised in my initial review, and their clarifications have improved my understanding and appreciation of the work. As a result, I have decided to raise my score.

**Limitations:**

yes, at the end of *Conclusion* section.

**Paper Formatting Concerns:**

There are no formatting concerns.

**Quality:**

3

**Strengths And Weaknesses:**

# Strength

1. The proposed linear alignment measurement is simple but effective and efficient (linear complexity) to incorporate the pair-wise similarities of missing data.

2. The proposed IMDC-DIA achieves promising clustering results compared to the existing methods.

3. The limitations are discussed at the end of *Conclusion* section.

# Weaknesses

1. The authors are expected to doubly check the Equation (1). This paper argues that the latent representation is unique to all data views. But $z_i^{(v)}$ in Equation (1) indicates that each data view corresponds a latent representation. The authors are encouraged to clarify this ambiguity and revise the notation accordingly. It is also recommended to carefully proofread the entire manuscript to eliminate typos and inconsistencies.

2. Although the paper reports accuracy and normalized mutual information (NMI), purity is another standard metric in clustering evaluation. Including purity results (if available) would provide a more comprehensive performance comparison.

3. In ablation study, "Three settings are designed ...". However, only two settings are designed and tested.

4. The convergence should be analyzed empirically at least.

5. The paper focuses on data imputation as a core component, yet recent related works on imputation for incomplete multi-view clustering are not cited, e.g., Incomplete Multi-view Clustering via Prototype-based Imputation, IJCAI 2023, Decoupled Contrastive Multi-view Clustering with High-order Random Walks, AAAI 2024.

6. Theorem 1 plays a central role in the paper’s theoretical foundation, but its statement lacks clarity. Specifically, key terms such as "consistency" and "pair-wise linear similarities" are not formally defined, making the theorem difficult to interpret or evaluate.

---

> ### Author Rebuttal · Authors · 2025-07-30
>
> Thanks so much for your recognition on the proposed linear alignment measurement and the effectiveness of experiment results!
>
> **Q1. Typos and inconsistencies**
>
> Thanks. So sorry for the typos! $\mathbf{z}_ i^{(v)}$ should be revised to $\mathbf{z}_ i$, where Eq. (1) should be
> $$\hat{\mathbf{x}}_i^{(v)} = g _{\Theta_v}(\mathbf{z}_i).$$
> Also, we have proofread the whole manuscript to eliminate all typos and inconsistencies.
> Specifically, we have added the bracket of $v$ in Eq. (3) to keep consistency of all equations, i.e.
> $$\bar{\mathbf{x}}_i^{(v)} = \begin{cases}
>     \mathbf{x}_i^{(v)} & \text{ if } \mathbf{a} _{i,v} = 1, \\\\
>     \hat{\mathbf{x}}_i^{(v)} & \text{ if } \mathbf{a} _{i,v} = 0.
>   \end{cases}$$
>
> **Q2. Purity results**
>
> Thanks. After running the experiments again, we have obtained not only the purity results but also the ARI results additionally. In the following are those on Handwritten dataset, while the others will be provided in Appendix. It can be seen that the purity and ARI results follow the same trend with the ACC and NMI results.
>
> | Metric |  | PUR |  |  |  |  |  | ARI |  |  |  |  |  |
> |---|---|:---:|:---:|:---:|:---:|:---:|:---:|:---:|:---:|:---:|:---:|:---:|:---:|
> | Missing ratio |  | 0.1 | 0.3 | 0.5 | 0.7 | 0.9 | avg. | 0.1 | 0.3 | 0.5 | 0.7 | 0.9 | avg. |
> | HandWritten | DITA-IMVC | 77.85 | 80.90 | 81.38 | 81.02 | 55.00 | 75.23 | 66.61 | 70.78 | 70.87 | 68.67 | 35.92 | 62.57 |
> |  | DSIMVC | 81.13 | 80.73 | 79.88 | 77.50 | 51.18 | 74.08 | 71.69 | 70.19 | 67.27 | 63.91 | 32.36 | 61.08 |
> |  | DCP | 83.97 | 78.08 | 79.40 | 73.40 | 13.42 | 65.65 | 75.32 | 61.27 | 62.61 | 53.62 | 0.02 | 50.57 |
> |  | DVIMC | 86.53 | 83.12 | 45.53 | 25.27 | 19.45 | 51.98 | 82.38 | 77.91 | 35.65 | 14.20 | 5.74 | 43.18 |
> |  | CPSPAN | 90.42 | 91.25 | 91.08 | 90.27 | 87.82 | 90.17 | 80.83 | 82.12 | 81.69 | 80.46 | 78.02 | 80.62 |
> |  | IMDC-DIA | 96.37 | 93.68 | 91.80 | 90.65 | 87.93 | 92.09 | 92.09 | 93.68 | 91.80 | 90.65 | 87.93 | 91.23 |
>
>
> | Metric |  |  | PUR |  |  |  |  |  | ARI |  |  |  |  |  |
> |---|---|---|:---:|:---:|:---:|:---:|:---:|:---:|:---:|:---:|:---:|:---:|:---:|:---:|
> | Missing ratio |  |  | 0.1 | 0.3 | 0.5 | 0.7 | 0.9 | avg. | 0.1 | 0.3 | 0.5 | 0.7 | 0.9 | avg. |
> | HandWritten | w/o imp. | avg | 78.12 | 68.13 | 59.27 | 53.78 | 44.48 | 60.76 | 62.70 | 44.29 | 24.49 | 18.14 | 11.49 | 32.22 |
> |  |  | zero | 83.98 | 70.98 | 62.73 | 51.35 | 46.63 | 63.14 | 71.39 | 47.75 | 32.75 | 21.58 | 15.48 | 37.79 |
> |  |  | rand | 80.18 | 71.13 | 59.95 | 55.62 | 46.58 | 62.69 | 70.79 | 55.61 | 32.89 | 25.63 | 15.86 | 40.16 |
> |  |  | none | 96.13 | 84.73 | 89.28 | 79.67 | 56.68 | 81.30 | 91.59 | 77.60 | 80.11 | 69.16 | 37.18 | 71.13 |
> |  | w/o align. |  | 93.62 | 88.18 | 90.57 | 83.77 | 84.12 | 88.05 | 86.73 | 78.23 | 80.55 | 71.71 | 69.92 | 77.43 |
> |  | IMDC-DIA |  | 96.37 | 93.68 | 91.80 | 90.65 | 87.93 | 92.09 | 92.09 | 86.37 | 82.60 | 80.53 | 75.42 | 83.40 |
>
>
> **Q3. Three settings**
>
> Thanks. Sorry for the inconsistency. In ablation study, three settings are considered, i.e.
> 1. The *avg.*, *zero* and *rand* of *w/o imp.* substitute the data imputation into completing the missing data with averages of available data observations, zeros and random values, respectively.
> 2. The *none* of *w/o imp.* refers to only using the available data observations in data alignment module rather than completing the missing.
> 3. *w/o align.* removes the data alignment module, which can be easily implemented by setting parameter $\beta$ to $0$.
>
> **Q4. Empirical convergence analysis**
>
> Thanks. We have recorded the loss value at each training epoch. Since the figures cannot be uploaded in Rebuttal period, we list the loss values on HandWritten dataset at each 40 epochs in the following:
>
> `
> 1.4718, 0.4360, 0.1024, 0.0549, 0.0281, -0.0070, -0.0433, -0.0717, -0.0925, -0.1082, -0.1203, -0.1298, -0.1375, -0.1437, -0.1489, -0.1534, -0.1573, -0.1607, -0.1637, -0.1663, -0.1687, -0.1708, -0.1726, -0.1743, -0.1757, -0.1769, -0.1780, -0.1790, -0.1799, -0.1807, -0.1814, -0.1821, -0.1828, -0.1834, -0.1839, -0.1845, -0.1850, -0.1854, -0.1859, -0.1863, -0.1867, -0.1870, -0.1874, -0.1877, -0.1880, -0.1883, -0.1886, -0.1888, -0.1891, -0.1893
> `
>
> It can be observed that the loss value continuously decreases and finally converges to the minimum. In next version, we will plot them and provide the loss value variation on all datasets in Appendix.
>
> **Q5. Related work**
>
> Thanks. The two researches present classical incomplete multi-view clustering methods. Specifically,
> - The first one proposes a novel dual-stream model by performing data recovery using the prototypes in the missing view and the sample-prototype relationship inherited from the observed view.
> - The second one proposes a novel robust method by leveraging random walks to progressively identify data pairs globally and performing inter-view and intra-view contrastive learning in different embedding spaces.
>
> We will introduce them in the Related Work section of next version.
>
> **Q6. Clarity of Theorem 1**
>
> Thanks. Sorry for the missing of these two concepts! They are introduced as follows. Also, we polish Theorem 1 and its proof carefully.
>
>
> Denoting vector $\mathbf{x}_ i$ to the $i$-th row of an arbitrary matrix $\mathbf{X}$, the **pair-wise linear similarity** between its $i$-th and $j$-th rows is measured by their dot-products that
> \begin{equation}
>   k_{i,j} = \mathbf{x}_i\mathbf{x}_j^\top, \quad w.r.t.\quad i, j\in \{1, 2, \cdots, N\}
> \end{equation}
>
> With considering two arbitrary matrices $\mathbf{X}_ 1$ and $\mathbf{X}_ 2$ coherently, their pair-wise linear similarities should be
> \begin{equation}
>   k^1_ {i,j} = \mathbf{x}^1_ i\mathbf{x}^{1\top}_ j \quad \text{and} \quad k^2_ {i,j} = \mathbf{x}^2_ i\mathbf{x}^{2\top}_ j
> \end{equation}
> On this basis, the similarity **consistency** of these two matrices can be measured by the sum of their dot-products as
> \begin{equation}
>   c = \frac{\sum_ {i,j=1}^N k^1_ {i,j}k^2_ {i,j}}{\sqrt{\sum_{i,j=1}^N k^1_ {i,j}k^1_ {i,j}}\sqrt{\sum_ {i,j=1}^N k^2_ {i,j}k^2_ {i,j}}}
> \end{equation}
> where the denominator is a scaler term to ensure the obtained consistency in range $[-1, 1]$.
>
> **Theorem 1**. *The linear alignment of two arbitrary matrices measures the consistency of the pair-wise linear similarities of their rows.*
>
> **Proof**.
> The consistency can be transformed to
> $$
> c  = \frac{\sum_ {i,j=1}^N k^1_ {i,j}k^2_ {i,j}}{\sqrt{\sum_ {i,j=1}^N k^1_ {i,j}k^1_ {i,j}}\sqrt{\sum_ {i,j=1}^N k^2_ {i,j}k^2_ {i,j}}} = \frac{\sum_ {i,j=1}^N k^1_ {i,j}k^2_ {j,i}}{\sqrt{\sum_ {i,j=1}^N k^1_ {i,j}k^1_ {i,j}}\sqrt{\sum_ {i,j=1}^N k^2_ {i,j}k^2_ {i,j}}}
> $$
> $$
>  = \frac{<\mathbf{K}_1, \mathbf{K}_2>_F}{\sqrt{<\mathbf{K}_1, \mathbf{K}_1>_F<\mathbf{K}_2, \mathbf{K}_2>_F}} = \frac{\mathrm{Tr}[(\mathbf{X}_1\mathbf{X}_1^\top)(\mathbf{X}_2\mathbf{X}_2^\top)]}{\sqrt{\mathrm{Tr}[(\mathbf{X}_1\mathbf{X}_1^\top)(\mathbf{X}_1\mathbf{X}_1^\top)]\cdot\mathrm{Tr}[(\mathbf{X}_2\mathbf{X}_2^\top)(\mathbf{X}_2\mathbf{X}_2^\top)]}}
> $$
> $$
> = \frac{\mathrm{Tr}[(\mathbf{X}_1^\top\mathbf{X}_2)^\top(\mathbf{X}_1^\top\mathbf{X}_2)]}{\sqrt{\mathrm{Tr}[(\mathbf{X}_1^\top\mathbf{X}_1)(\mathbf{X}_1^\top\mathbf{X}_1)]\cdot\mathrm{Tr}[(\mathbf{X}_2^\top\mathbf{X}_2)(\mathbf{X}_2^\top\mathbf{X}_2)]}} = \frac{\|\mathbf{X}_1^\top\mathbf{X}_2\|_F^2}{\|\mathbf{X}_1^\top\mathbf{X}_1\|_F\cdot\|\mathbf{X}_2^\top\mathbf{X}_2\|_F} = LA,
> $$
>
>   in which the second equation holds for
>   \begin{equation}
>     k^2_ {j,i} = \mathbf{x}_ {2,j}\mathbf{x}_ {2,i}^\top = \mathbf{x}_ {2,i}\mathbf{x}_ {2,j}^\top = k^2_ {i,j},
>   \end{equation}
>
>   while the third holds for
>
> \begin{equation}
>  \mathbf{K}_ 1 = \begin{bmatrix}
>     \mathbf{x}_ {1,1}\mathbf{x}_ {1,1}^\top & \mathbf{x}_ {1,1}\mathbf{x}_ {1,2}^\top  & \cdots & \mathbf{x}_ {1,1}\mathbf{x}_ {1,N}^\top \\\\
>     \mathbf{x}_ {1,2}\mathbf{x}_ {1,1}^\top & \mathbf{x}_ {1,2}\mathbf{x}_ {1,2}^\top  & \cdots & \mathbf{x}_ {1,2}\mathbf{x}_ {1,N}^\top \\\\
>     \vdots & \vdots  & \ddots & \vdots \\\\
>     \mathbf{x}_ {1,N}\mathbf{x}_ {1,1}^\top & \mathbf{x}_ {1,N}\mathbf{x}_ {1,2}^\top  & \cdots & \mathbf{x}_ {1,N}\mathbf{x}_ {1,N}^\top \\\\
>   \end{bmatrix} = \begin{bmatrix}
>     k^1_ {1,1} & k^1_ {1,2} & \cdots & k^1_ {1,3} \\\\
>     k^1_ {2,2} & k^1_ {2,2} & \cdots & k^1_ {2,3} \\\\
>     \cdots & \cdots & \ddots & \cdots \\\\
>     k^1_ {N,1} & k^1_ {N,2} & \cdots & k^1_ {N,N}
>   \end{bmatrix}
> \end{equation}
>
> and
>
> $$
> \mathbf{K}_ 2 = \begin{bmatrix}
>     \mathbf{x}_ {2,1}\mathbf{x}_ {2,1}^\top & \mathbf{x}_ {2,1}\mathbf{x}_ {2,2}^\top  & \cdots & \mathbf{x}_ {2,1}\mathbf{x}_ {2,N}^\top \\\\
>     \mathbf{x}_ {2,2}\mathbf{x}_ {2,1}^\top & \mathbf{x}_ {2,2}\mathbf{x}_ {2,2}^\top  & \cdots & \mathbf{x}_ {2,2}\mathbf{x}_ {2,N}^\top \\\\
>     \vdots & \vdots  & \ddots & \vdots \\\\
>     \mathbf{x}_ {2,N}\mathbf{x}_ {2,1}^\top & \mathbf{x}_ {2,N}\mathbf{x}_ {2,2}^\top  & \cdots & \mathbf{x}_ {2,N}\mathbf{x}_ {2,N}^\top \\\\
>   \end{bmatrix} = \begin{bmatrix}
>     k^2_ {1,1} & k^2_ {1,2} & \cdots & k^2_ {1,3} \\\\
>     k^2_ {2,2} & k^2_ {2,2} & \cdots & k^2_ {2,3} \\\\
>     \cdots & \cdots & \ddots & \cdots \\\\
>     k^2_ {N,1} & k^2_ {N,2} & \cdots & k^2_ {N,N}
>   \end{bmatrix}
> $$
>
> In summary, $c = LA$, illustrating that the linear alignment of two arbitrary matrices equals to the consistency of the pair-wise linear similarities of their rows.
> This completes the proof.

---

> > ### Comment · Reviewer_aouo · 2025-08-02
> >
> > I thank the authors for their detailed rebuttal. I believe they have adequately addressed the main concerns I raised in my initial review, and their clarifications have improved my understanding and appreciation of the work. As a result, I have decided to raise my score.

---

> > > ### Author Response · Authors · 2025-08-06
> > > **The response**
> > >
> > > Thanks so much for your recognition!

---

### Decision · Program_Chairs · 2025-09-17

**Decision:**

Accept (poster)

**Comment:**

The manuscript proposes an incomplete multi-view deep clustering method with data imputation and alignment.
Reviewers agree that the proposed approach is novel, the motivation and technical details are well-written, and the overall design of experiments including ablation study is thorough.
While the original manuscript had an issue in providing some experimental results and complete theoretical statements, they have been solved through the author-reviewer discussion.